**Data Availability Statement:** All relevant data are within the manuscript and its Supporting Information files.

# Colonization of the *Caenorhabditis elegans* gut with human enteric bacterial pathogens leads to proteostasis disruption that is rescued by butyrate

Alyssa C. Walker [1], Rohan Bhargava [1], Alfonso S. Vaziriyan-Sani [1], Christine Pourciau [1], Emily T. Donahue [1], Autumn S. Dove [1], Michael J. Gebhardt [2], Garrett L. Ellward [1], Tony Romeo [1], Daniel M. Czyż [1]*

**1** Department of Microbiology & Cell Science, University of Florida, Gainesville, Florida, United States of America, **2** Division of Infectious Diseases, Boston Children's Hospital, Harvard Medical School, Boston, Massachusetts, United States of America

* dczyz@ufl.edu

## Abstract

Protein conformational diseases are characterized by misfolding and toxic aggregation of metastable proteins, often culminating in neurodegeneration. Enteric bacteria influence the pathogenesis of neurodegenerative diseases; however, the complexity of the human microbiome hinders our understanding of how individual microbes influence these diseases. Disruption of host protein homeostasis, or proteostasis, affects the onset and progression of these diseases. To investigate the effect of bacteria on host proteostasis, we used *Caenorhabditis elegans* expressing tissue-specific polyglutamine reporters that detect changes in the protein folding environment. We found that colonization of the *C. elegans* gut with enteric bacterial pathogens disrupted proteostasis in the intestine, muscle, neurons, and the gonad, while the presence of bacteria that conditionally synthesize butyrate, a molecule previously shown to be beneficial in neurodegenerative disease models, suppressed aggregation and the associated proteotoxicity. Co-colonization with this butyrogenic strain suppressed bacteria-induced protein aggregation, emphasizing the importance of microbial interaction and its impact on host proteostasis. Further experiments demonstrated that the beneficial effect of butyrate depended on the bacteria that colonized the gut and that this protective effect required SKN-1/Nrf2 and DAF-16/FOXO transcription factors. We also found that bacteria-derived protein aggregates contribute to the observed disruption of host proteostasis. Together, these results reveal the significance of enteric infection and gut dysbiosis on the pathogenesis of protein conformational diseases and demonstrate the potential of using butyrate-producing microbes as a preventative and treatment strategy for neurodegenerative disease.

**Funding:** This work was supported in part by the Infectious Diseases Society of America and Start-up Funding provided by the Microbiology & Cell Science Department at the University of Florida Institute of Food and Agricultural Sciences to DMC. The funders had no role in study design, data collection and analysis, decision to publish, or preparation of the manuscript.

**Competing interests:** The authors have declared that no competing interests exist.

## Author summary

Protein conformational diseases are one of the leading causes of geriatric death and disability, worldwide. Individuals suffering from these ailments are limited to palliative care, as there are no cures or effective treatments. Correlational evidence suggests that the human gut microbiota is a culprit, but the effect of individual bacteria remains elusive, in part, due to the complexity of the microbiome. A single-bacterium approach can help to deconvolute the complexity of the microbiome and reveal the effect of individual bacterial species on organismal proteostasis. As such, we utilized the intestine of *C. elegans* as a "test tube" to identify the effect of bacteria on the host using tissue-specific polyglutamine repeats as protein folding sensors. We found that colonization of the *C. elegans* intestine with pathogenic gram-negative bacteria disrupted proteostasis in the intestine, muscle, neurons, and gonads. Furthermore, we demonstrated that butyrogenic bacteria enhanced proteostasis, which was evidenced by a decrease in polyglutamine aggregation and suppression of aggregate-dependent toxicity. Further experiments revealed that co-colonization with butyrogenic bacteria inhibited protein aggregation in *C. elegans* and the butyrate-mediated suppression of aggregation is dependent on SKN-1/Nrf2 and DAF-16/ FOXO–two transcription factors involved in the regulation of oxidative stress responses. While the mechanism of bacteria-mediated induction of protein aggregation remains elusive, our results suggest that bacterial aggregates, in addition to the contribution of oxidative stress, are the contributing factor. These results are intriguing as they suggest that enteric bacteria directly contribute to the pathogenicity of protein conformational diseases.

## Introduction

Neurodegenerative protein conformational diseases (PCDs), including amyotrophic lateral sclerosis (ALS), Alzheimer's, Huntington's, and Parkinson's disease, are characterized by the misfolding and aggregation of metastable proteins that reside within the proteome, often resulting in loss of tissue function that manifests in disease progression [1]. Despite the high prevalence and enormous financial and social burdens imposed on afflicted individuals and their families [2], no effective treatment or cure has been found; moreover, the etiology of these diseases remains largely unknown [3]. Factors such as age, diet, stress, trauma, toxins, infections, or antibiotics, have been shown to increase the risk of PCDs [4–10]. Notably, these triggers are also associated with changes in the microbiome [11–13], suggesting that bacteria may contribute to the pathogenesis of PCDs, which may explain their sporadic onset. None-theless, the relationship between the microbiome and disease progression remains poorly defined.

The human gut microbiota (HGM) is a complex assembly of microorganisms capable of synthesizing molecules that can impact the host's physiology [14]. A healthy commensal relationship between a balanced microbiome and the host is fostered by their cross-talk [15]. Conversely, gut dysbiosis results from the loss of beneficial bacteria accompanied by the overgrowth of opportunistic pathogens, a condition often induced by antibiotics [16,17]. Recent evidence has established a link between the HGM and PCDs, whereby gut dysbiosis or direct enteric infection exacerbate the disease [18–20]. Commensal residents of the human microbiota, specifically those involved in the synthesis of short-chain fatty acids (SCFAs) such as butyrate, were shown to be beneficial to the host [21]. Unfortunately, environmental factors and the complexity of the human microbiome often hinder the consistency of results from

experiments that pertain to PCDs [22]. To eliminate such complexity, we employed *Caenorhabditis elegans* as a model to study the effect of enteric pathogens on host proteostasis and examined the benefits provided by butyrogenic bacteria.

Here, we show that colonization of the *C. elegans* gut with gram-negative enteric bacterial pathogens disrupts host proteostasis, leading to aggregation and proteotoxicity of polyglutamine (polyQ) tracts across multiple tissues, including intestine, muscle, neurons, and the gonads. Such tissue non-autonomous effects of bacteria on the host may explain the impact of the gut microbiota on the onset and progression of PCDs. While pathogenic bacteria contribute to aggregation, commensal strains suppress it. Additionally, we show that butyrate and butyrogenic bacteria enhance host proteostasis and lead to suppression of bacteria-mediated polyQ aggregation and proteotoxicity. Oxidative stress and bacteria-derived protein aggregates seem to contribute to the observed disruption of proteostasis. Collectively, our results suggest that dysbiosis between enteric pathogens and commensal butyrogenic bacteria contributes to the pathogenicity of PCDs.

## Results

### Colonization of the *C. elegans* intestine with human bacterial pathogens disrupts proteostasis and affects animal motility

We tested the ability of select bacterial species from the *Enterobacteriaceae* family to affect the protein folding environment in *C. elegans* upon intestinal colonization. Select bacterial species were from the following genera: *Escherichia*, *Klebsiella*, *Proteus*, *Citrobacter*, *Shigella*, and *Salmonella*, as well as additional pathogenic bacteria that are associated with gut microbiota; these include gram-negative *Pseudomonas* and *Acinetobacter*. Although the physical manifestations of human proteopathies involve aggregate-induced dysfunction of neuronal and musculoskeletal tissue, we concluded that the intestinal tissue was best suited for our initial experiments because it is the immediate environment of bacterial colonization, presumably resulting in the most robust effect. As such, to assess the effect of these candidate strains on the protein folding environment in the *C. elegans* gut, we used animals that constitutively express intestine-specific polyQs fused to yellow fluorescent protein (polyQ44::YFP) [23]. Nematodes carrying this reporter exhibit age-dependent aggregation of intestinal polyQs (**Fig 1A**). These aggregates present as quantifiable fluorescent foci and were previously characterized as a proxy to assess the influence of bacteria on aggregation [23]. To determine the age at which nematodes harbor an intermediate range of aggregation for identifying bacteria that suppress or enhance aggregation, we quantified aggregates between days 1–5, post-hatching. Worms were grown at room temperature (~23˚C) on the control bacteria, *E. coli* OP50. Because a high number of aggregates can introduce counting error, an aggregation threshold was set at 40 aggregates per intestine to ensure the reliability of counting. Aggregates increased each day, becoming visible after three days, and reaching our set threshold after five days (**Fig 1A**). Hence, to identify bacteria that either enhance or suppress aggregation, we chose to grow *C. elegans* on test bacteria for four days before assessing the ability of each strain to influence protein aggregation in the intestine. Out of 19 strains tested, one did not have any effect on aggregation, six strains significantly enhanced aggregation two- to three-fold, and 12 strains enhanced aggregation more than three-fold with respect to *E. coli* OP50 (**Fig 1B**). We found that the strongest enhancers were *P. aeruginosa*, *K. pneumoniae*, *C. freundii*, *S.* Typhimurium, *K. aerogenes*, and *P. mirabilis*. The only strain that did not significantly induce aggregation relative to *E. coli* OP50 was *E. coli* HB101, a non-pathogenic strain derived from *E. coli* K-12. To eliminate the possibility that bacteria change polyQ expression levels, using an antibody specific to YFP/GFP (**S1A Fig**), we confirmed that polyQ44:YFP soluble protein levels do not

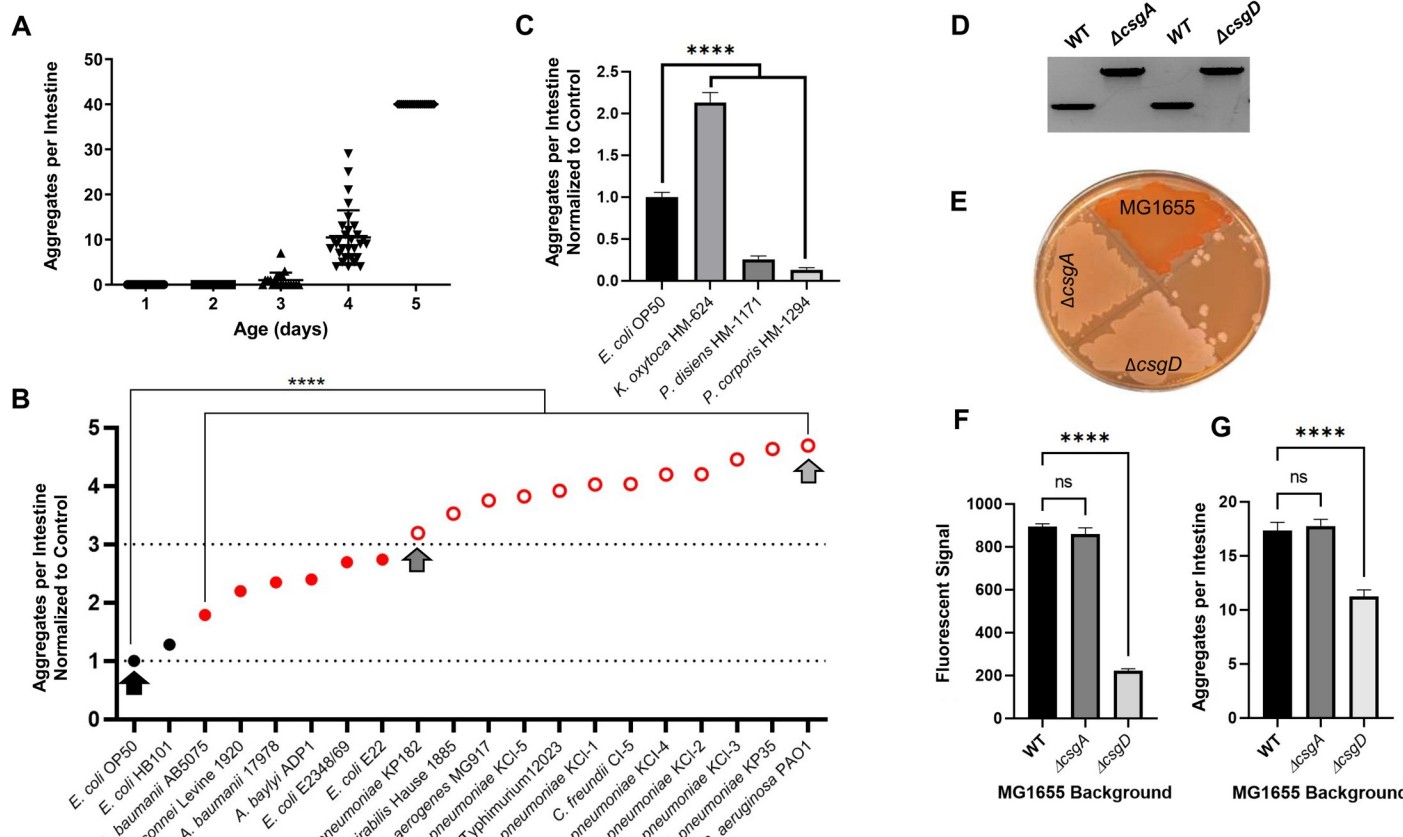

**Fig 1. Gram-negative bacteria found in the human gut affect *C. elegans* proteostasis and enhance protein aggregation in the intestine. A)** Age-dependent aggregation of intestinal polyQs (polyQ44) colonized by *E. coli* OP50. Data are represented as the number of aggregates per 20–30 intestines quantified in animals at day 1–5 post-hatching. **B)** The effect of select enteric bacteria on polyQ aggregation in the *C. elegans* intestine. Data are represented as the average number of aggregates per *C. elegans* intestine normalized to *E. coli* OP50 control strain (lower dotted line). Each data point is an average of a minimum of three independent experiments with a total of at least 100 animals. Black circles represent control bacteria and bacteria that did not have any significant effect on polyglutamine aggregation upon colonization of the *C. elegans* intestine. Red solid circles represent bacteria that significantly enhanced aggregation by <3-fold. Red open circles represent bacteria that significantly enhanced aggregation by >3-fold (arbitrary threshold). Arrows represent bacteria that were chosen for follow-up experiments. **C)** The effect of select commensal bacteria on polyQ aggregation in the *C. elegans* intestine. Data are represented as the average number of aggregates per *C. elegans* intestine normalized to *E. coli* OP50 control strain. Each data point is an average of a minimum of three independent experiments with a total of at least 100 animals. **D)** PCR confirmation of *E. coli* MG1655 curli mutant strains. WT, *csgA*::*kan* (left two bands) amplified with primers flanking the Δ*csgA* locus. WT, *csgD*::*kan* (right two bans) amplified with primers flanking the Δ*csgD* locus. **E)** Phenotypic confirmation of the curli-deficient Δ*csgA* and Δ*csgD* mutant strains using Congo Red plate assay. **F)** ProteoStat staining of total aggregates produced by *E. coli* MG1655 wild-type (WT), MG1655 Δ*csgA*, and MG1655 Δ*csgD*. Data are represented as the average fluorescent signal per bacterial strain stained with ProteoStat. Each data point is an average of two independent experiments with three replicates per run. **G)** The effect of *E. coli* MG1655 curli mutants on polyQ (polyQ44) aggregation in the *C. elegans* intestine. Data are represented as the average number of aggregates per *C. elegans* intestine. Each data point is an average of three independent experiments with a total of 90 animals. Error bars represent standard error of the mean (SEM). Statistical significance was calculated using one-way analysis of variance (ANOVA) followed by multiple comparison Dunnett's post-hoc test (****p< 0.0001).

change between *E. coli* OP50 and *P. aeruginosa* PAO1 –two bacterial strains that elicited the lowest and highest intestinal aggregation, respectively (**S1B Fig**). Moreover, western blot analysis of the insoluble polyQ44::YFP fraction confirmed bacteria-mediated increase in aggregation (**S2A Fig**), as it parallels the aggregation profile of worms fed *E. coli* OP50 and *P. aeruginosa* PAO1 (**Fig 1B**). These results are intriguing as they demonstrate that common human enteric pathogens significantly affect polyQ aggregation, and therefore host proteostasis. We next asked whether gram-negative bacteria that are associated with commensal microflora would also enhance aggregation. We tested three strains known to be part of the commensal microbiome: *K. oxytoca* HM-624, *Prevotella disiens* HM-1171, and *Prevotella*

*corporis* HM-1294. While *K. oxytoca* enhanced aggregation, as we have seen with the other *Klebsiella* spp., both *Prevotella* spp. suppressed aggregation relative to *E. coli* OP50 (**Fig 1C**). In fact, *P. corporis* almost completely suppressed aggregation.

To begin understanding how bacteria could contribute to the disruption of host proteostasis, we used *E. coli* defective in the production of curli, which are aggregation-prone amyloids that are specific to *Enterobacteriaceae* [24]. We generated two mutant *E. coli* strains: one deficient in *csgA*, which encodes an amyloid structural subunit of curli fimbriae, and one deficient in *csgD*, a transcription factor that positively regulates the curli operon (**Fig 1D**). While staining with congo red confirmed a deficiency in curli production (**Fig 1E**), staining for bacterial protein aggregates using ProteoStat, which provides superior sensitivity in aggregate detection [25], revealed that only *csgD* mutant cells have significantly fewer aggregates (**Fig 1F**). In agreement with these results, we found that colonization of *C. elegans* only with the Δ*csgD E. coli* strain led to a significant decrease in the number of polyQ aggregates per intestine (**Fig 1G**). These results suggest that bacteria-derived aggregates significantly contribute to the disruption of host proteostasis.

Tissue non-autonomous effects between *C. elegans* neurons and other somatic cells (i.e., intestine and epithelium) have been recently described where perturbation of proteostasis in one tissue impacts another [26–28]. As such, we postulated that bacteria-induced disruption of proteostasis in the intestine could affect other tissues. To begin assessing the impact of bacteria on muscle, we examined motility. The nematodes that express intestinal polyQ have an integrated *rol-6* marker, which results in a roller phenotype and renders animals incompatible with known motility assays, such as thrashing. Thus, we developed a novel motility readout that entails sliding an eyebrow hair under the mid-section of the worm and counting the number of seconds until the worm completely crawls off—we refer to this phenotype as **t**ime-**o**ff-**p**ick (TOP) (**Fig 2A**). We found that TOP increases with age and the decrease in motility is dependent on polyQ when compared to control roller worms that do not express polyQs, as well as those that express non-aggregating polyQs (**Fig 2B**). To ensure that the TOP method measures polyQ-specific defects and not an undesired effect related to the roller phenotype or bacterial pathogenicity, we used non-roller strains that express muscle-specific polyQ35 (*unc-54p*::Q35::YFP, AM140) and polyQ40 (*unc-54p*::Q40::YFP, AM141). Both of these strains are known to exhibit polyQ-length and age-dependent motility defects [29]. In support of the functionality of our assay, we found that worms expressing these constructs also exhibited a

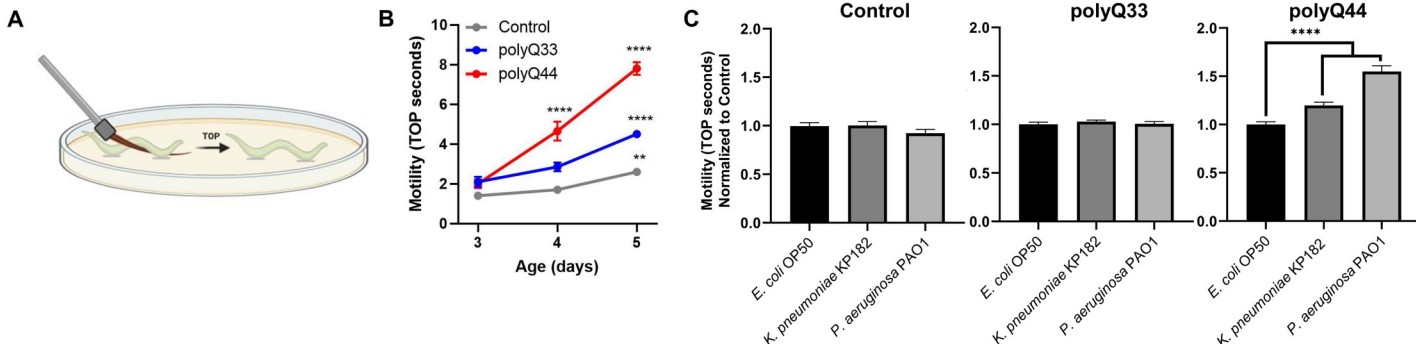

**Fig 2. Colonization of *C. elegans* expressing intestinal polyQ with human enteric pathogens influences motility. A)** A cartoon depicting **t**ime-**o**ff-**p**ick (TOP) phenotype measure as the time (seconds) it takes a worm to crawl off a pick when pickup up by the midbody section. **B)** Age-dependent changes of motility in animals expressing intestinal polyQ (polyQ33 and polyQ44) grown on *E. coli* OP50 control strain. Higher motility measured in TOP seconds indicates a higher motility defect. Data are represented as the average TOP per worm. Each data point represents 20 worms. **C)** Motility of animals expressing intestinal polyQ grown on *K. pneumoniae* KP182 and *P. aeruginosa* PAO1. Data are represented as the average TOP per 60 worms over three independent experiments, normalized to *E. coli* OP50 control strain. Error bars represent SEM. Statistical significance was calculated using one-way ANOVA followed by multiple comparison Dunnett's post-hoc test (**p<0.01, ****p <0.0001).

significant polyQ-length and age-dependent TOP motility defect that correlated with the enumeration of body bends, which is a well-established assessment of motility in *C. elegans* (**S3A Fig**). Additionally, colonization of these polyQ-expressing strains with *P. aeruginosa* PAO1 inhibited motility in both assays yet had little to no effect in wild-type N2 animals. These findings indicate that polyQ toxicity, and not bacterial pathogenicity, is the factor that is measured by TOP (**S3B Fig**). Therefore, we further used the TOP method to determine whether colonization of the *C. elegans* intestine with bacteria that induced polyQ aggregation aggravates this phenotype. We chose two bacterial strains: *K. pneumoniae* KP182 (medium inducer); *P. aeruginosa* PAO1 (high inducer) and found that the "time-off-pick" was significantly increased by both strains (**Fig 2C**), supporting the trend seen in their respective aggregation profiles (**Fig 1B**). We did not detect any effect on the TOP phenotype in roller controls or in roller nematodes expressing a non-aggregating and intestine-specific polyQ33, confirming that this motility defect was the result of bacteria-induced polyQ-dependent toxicity rather than general pathogenicity from a bacterial infection (**Fig 2C**). While it is known that worms expressing polyQ33 do not exhibit age-dependent aggregation, we wanted to ensure that aggregation does not occur when these animals are colonized by bacteria that induce polyQ44 aggregation. We did not detect polyQ33 aggregates in animals colonized with the strongest inducer, *P. aeruginosa* PAO1 (**S4 Fig**). Based on our results, we conclude that bacterial colonization of the *C. elegans* intestine affects protein folding in the immediate environment and that such localized intestinal protein aggregation influences motility, likely by acting on muscle or neuronal tissues.

## Bacterial colonization of the *C. elegans* intestine disrupts protein folding in distal tissues

**Muscle tissue.** Because bacteria-induced polyQ aggregation in the intestine influenced *C. elegans* motility, an effect that likely involves distal tissues, we sought to determine whether the influence of bacteria on protein aggregation is confined to the intestine or if bacteria can impact protein folding across other tissues. We employed nematodes expressing muscle-specific polyQ35 (*unc-54p*::Q35::YFP) as a proxy to detect changes in proteostasis in that tissue. Different tissues have different thresholds for polyQ aggregation; therefore, while polyQ33 did not aggregate in the intestine, polyQ35 aggregates in the muscle. This reporter exhibits age-dependent and muscle-specific aggregation and the associated motility defect [29]. We colonized the intestines of these worms with two strains that enhanced intestinal polyQ aggregation: *K. pneumoniae* KP182 and *P. aeruginosa* PAO1. Animals expressing muscle polyQs exhibit a more robust aggregation profile compared to the intestinal polyQ model; thus, we had to quantify the aggregates on day three instead of day four. The results paralleled the trend seen in the aggregation profile of the intestinal polyQ model, where *K. pneumoniae* enhanced aggregation compared to control *E. coli* OP50 and *P. aeruginosa* PAO1 showed the most robust enhancement (**Fig 3A**). To determine whether the bacteria-induced proteostasis imbalance detected by the polyQ aggregation in the muscle also leads to tissue dysfunction, we employed a previously characterized motility readout to enumerate the body bends of polyQ-expressing worms fed these two bacterial strains. The animals expressing muscle polyQ exhibit a sharp and age-dependent decrease in body bends (**Fig 3B**). We found that colonization of the *C. elegans* intestine with enteric bacterial pathogens led to a significant defect in motility, particularly in worms fed the most potent enhancer of aggregation, *P. aeruginosa*, which decreased the number of body bends 2-fold when compared to the control *E. coli* OP50; a larger and more significant defect than what was observed in control wild-type animals (**Fig 3C and 3D**).

**Neuronal tissue.** To determine whether proteostasis in neuronal tissue is also affected by bacterial colonization of the intestine, we utilized worms carrying neuronal polyQ40

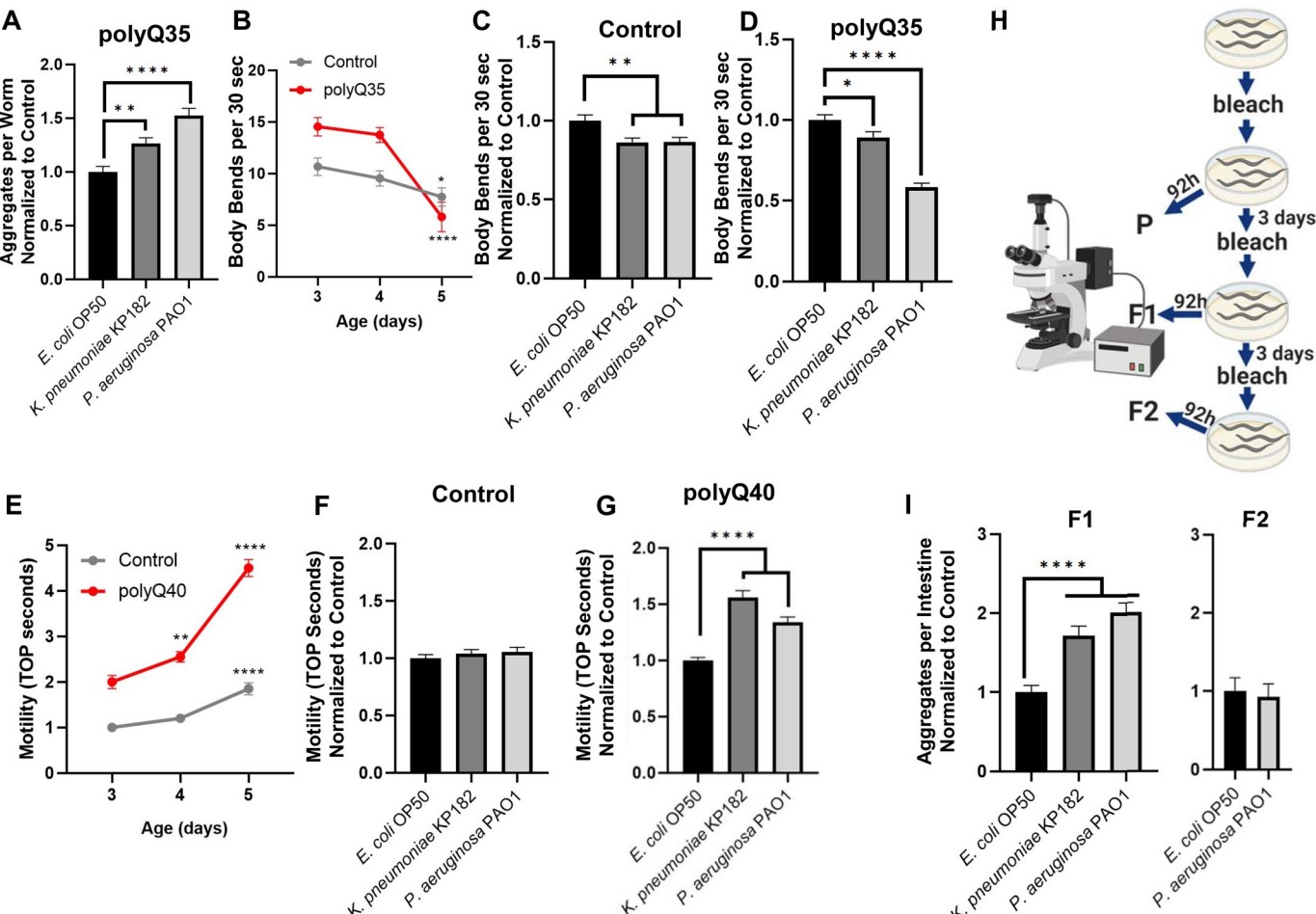

**Fig 3. Colonization of the *C. elegans* intestine with human enteric pathogens influences proteostasis in a tissue non-autonomous manner. A)** The effect of select bacteria on protein folding in the muscle. Data are represented as the average number of aggregates per worm normalized to *E. coli* OP50 control strain. Each bar is an average of three independent experiments with a total of 100 animals. **B)** Age-dependent decline in motility assessed by enumerating body bends per 30 seconds in muscle-specific polyQ35 and N2 control worms. The data are represented as the average number of body bends quantified on day 3–5 post-hatching in a total of 15 animals per each day. The effect of bacteria on the motility of **C)** control animals and **D)** animals expressing muscle-specific polyQ35. Data are represented as the average number of body bends per worm normalized to worms fed *E. coli* OP50 control strain. Each bar is an average of three independent experiments with a total of 45 animals. **E)** Age-dependent decline in motility assessed by increased TOP. The effect of bacteria on the motility of **F)** control animals and **G)** animals expressing neuron-specific polyQ40. Data are represented as the average TOP per worm normalized to animals fed *E. coli* OP50 control strain. Each bar is an average of three independent experiments with a total of 60 animals. **H)** A cartoon depicting experiments which demonstrated that bacterial colonization of the *C. elegans* intestine affects the F1 generation. **I)** Quantification of intestinal aggregates in the F1 and F2 generations from parental animals that were fed select test and control bacteria. Each bar represents the average number of aggregates per intestine normalized to the control (*E. coli* OP50). Data are representative of three (F1) and one (F2) independent experiments with a total of 100 and 30 animals, respectively. Error bars represent SEM. Statistical significance was calculated using one-way ANOVA followed by multiple comparison Dunnett's post-hoc test (*p<0.05, **p<0.01, ****p<0.0001).

(*F25B3.3p*::Q40::YFP) [30]. The small size and high concentration of neurons preclude us from accurately quantifying the exact number of aggregates in this tissue; as such, we measured toxicity by assessing TOP motility. Animals carrying this reporter exhibit an age-dependent decrease in motility compared to control N2 worms (**Fig 3E**). We found that colonization of the intestine with *K. pneumoniae* and *P. aeruginosa* led to a significant defect in motility when compared to the control *E. coli* OP50 (**Fig 3F and 3G**).

**Gonad.** Thus far, we demonstrated that colonization of the *C. elegans* gut with enteric bacterial pathogens leads to proteostasis imbalance and proteotoxicity in proximal tissues, including muscle and neurons. To further examine the extent to which gut bacteria influence host

proteostasis, we asked whether colonization of the *C. elegans* intestine had any effect on the gonad. If the colonization of the intestine affects the muscle and neurons, we reasoned that bacteria might also affect proteostasis in the germline—if this is the case, then we should be able to detect changes in polyQ aggregation in the progeny of parents that were colonized by pathogenic bacteria. To test the effect of bacteria on the next generation, we fed *C. elegans* either control *E. coli* OP50, *K. pneumoniae* KP182, or *P. aeruginosa* PAO1 for three days, followed by "bleaching" adults, and age-synchronizing the progeny. *C. elegans* synchronization involves dissolving gravid adults by hypochlorite treatment leaving intact and sterile embryos which hatch into L1 larvae. Synchronized F1 animals were placed on fresh plates containing a lawn of control *E. coli* OP50 and intestinal aggregates were enumerated after 92 hours (h). The F2 generation was isolated and assessed using the same method (**Fig 3H**). Strikingly, we found that colonization of the parental intestines with the select bacteria enhanced polyQ aggregation in the F1 progeny, despite the fact that they never encountered the bacteria that were fed to the parental generation (with the exception of *E. coli* OP50) (**S5 Fig**). The extent of the enhancement in the number of polyQ aggregates per worm correlated with previously assessed phenotypes where *P. aeruginosa* showed the highest induction of aggregation (**Fig 3I**). Since *P. aeruginosa* induced the strongest effect, we looked at the F2 progeny from a parental generation fed these bacteria. We saw no enhancement of aggregation compared to control *E. coli*, indicating that only F1 generation is affected (**Fig 3I**). Using western blotting, we ruled out the possibility that these results were due to changes in polyQ44::YFP expression level (**S1E Fig**) and we confirmed an increase of the insoluble polyQ44::YFP (**S2D Fig**). Collectively, these results are intriguing as they demonstrate that colonization of the parental intestines affects proteostasis in the offspring.

## Butyrate suppresses bacteria-induced polyglutamine aggregation and aggregate-dependent toxicity across tissues

The depletion of butyrogenic bacteria in the gut establishes an environment in which pathogenic bacteria, particularly *Enterobacteriaceae*, can flourish [31]. In addition, butyrate itself has been found to suppress the growth of pathogenic bacteria and was shown to protect against neurodegeneration [32–34]. To assess the effect of exogenous butyrate on the observed bacteria-induced aggregation, we first tested its effect on intestinal polyQ aggregation in *C. elegans*. We used all bacterial genera that induced protein aggregation by three-fold or more (**Fig 1B**). The physiological concentration of human colonic butyrate is estimated to be 10–20 mM, and a therapeutic dose of 150 mM butyrate was safely delivered via enema directly into the human large intestine [35,36]. Therefore, we tested the effect of butyrate on bacteria-induced protein aggregation at concentrations ranging from 10–100 mM. Unexpectedly, we found that lower concentrations of butyrate differentially affected host proteostasis where enhancement of aggregation was dependent on the type of bacteria colonizing the intestine: 10 mM: *S. enterica* 12023, *P. mirabilis*; 25 mM: *P. mirabilis*, *P. aeruginosa* PAO1; 50 mM: *P. aeruginosa* PAO1. However, at higher butyrate concentrations (>25 mM), bacteria-induced aggregation was suppressed in a dose-dependent manner in worms colonized with all bacteria except *P. aeruginosa*. Supplementation with 100 mM butyrate suppressed bacteria-induced aggregation across all strains. (**Fig 4**). These results are unexpected, as they suggest that at low physiological concentrations (<25 mM), butyrate may contribute to the bacteria-mediated disruption of host proteostasis, as demonstrated by differentially enhanced polyQ aggregation; however, with the exception of *P. aeruginosa* PAO1, higher concentrations (50–100 mM) of butyrate suppressed aggregation induced by all other bacterial strains, which suggests therapeutic potential. We employed western blot analysis of polyQ44::YFP levels to eliminate the possibility that these

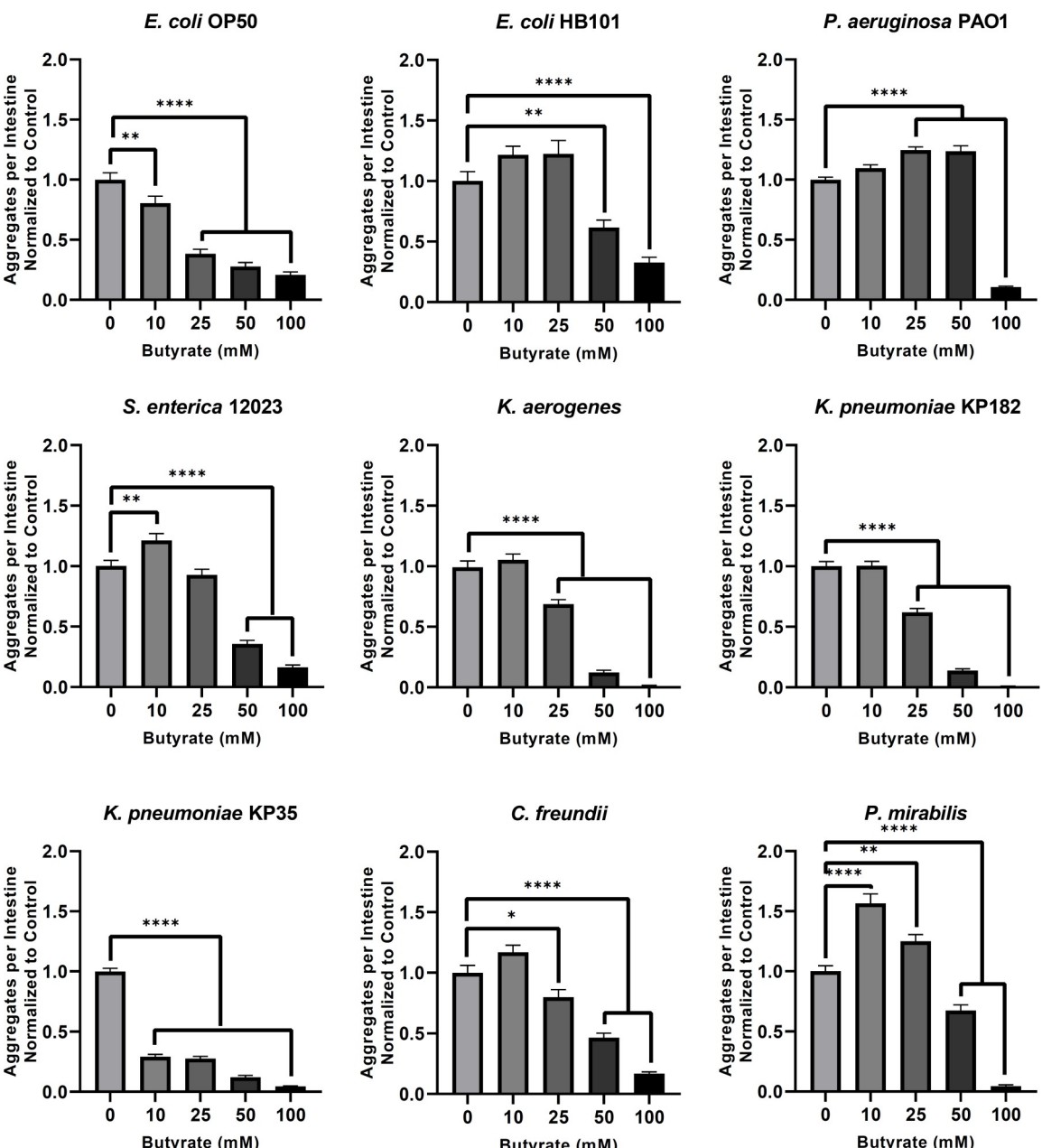

**Fig 4. The effect of butyrate on bacteria-induced aggregation in the intestine.** Data are represented as the average number of aggregates per intestine normalized to the control (0 mM butyrate). Each bar is an average of three independent experiments with a total of 100 animals. Error bars represent SEM. Statistical significance was calculated using one-way ANOVA followed by multiple comparison Dunnett's post-hoc test (*p<0.05, **p<0.01, ****p<0.0001).

results are due to changes in protein level (**S1C and S1D Fig**), and we confirmed these results by assessing insoluble fractions (**S2B and S2C Fig**).

Furthermore, to rule out the possibility that butyrate decreased bacteria-mediated protein aggregation by merely killing bacteria, we tested the effect of butyrate on bacterial growth and viability in liquid medium and on solid NGM plates, respectively, at concentrations up to 100 mM. While butyrate did affect the growth and viability of some bacteria, all of the strains tested remained viable. We found that most overnight cultures supplemented with butyrate at

concentrations that inhibited aggregation reached optical densities comparable to unsupplemented controls (**S6 Fig**). We also tested the effect of butyrate on bacteria cultured on solid NGM media and found that, while some strains were affected, all were viable across concentrations (**S7 Fig**). Additionally, feeding *C. elegans* dead bacteria resulted in higher aggregation profiles that are distinct from those treated with 100 mM butyrate, further suggesting that butyrate does not suppress bacteria-mediated aggregation by solely affecting bacterial viability (**S8 Fig**). Next, we asked whether butyrate could affect colonization of the intestine. We found that at 25 mM, butyrate significantly enhanced intestinal colonization by *E. coli* OP50, but exhibited no effect on colonization at 100 mM (**S9 Fig**). These results indicate that it is unlikely that the butyrate's effect on bacterial colonization affects aggregation.

A trade-off between organismal resources available to maintain either proteostasis or fecundity has been proposed [37]. As such, we asked whether butyrate affects fecundity in our experiments, which may explain the suppression of aggregation. Indeed, we found that butyrate decreases the number of progeny per worm, but the effect seems to depend on the presence of polyQ (**S10A and S10B Fig**). Moreover, 25 mM butyrate significantly decreased the number of progeny in worms colonized by *P. aeruginosa* (**S10C Fig**). Yet, we found that under the same conditions, butyrate increased polyQ aggregation in animals colonized by *P. aeruginosa* (**Figs 4** and S2C). These results indicate that a decrease in fecundity alone does not explain the observed suppression of polyQ aggregation.

To determine whether exogenous butyrate would also rescue the aggregate-dependent motility defect observed in the intestinal polyQ44 model, we measured the TOP phenotype in animals colonized by *E. coli* OP50, *K. pneumoniae* KP182, and *P. aeruginosa* PAO1 in the absence or presence of 100 mM butyrate. While little to no changes in TOP were observed in the roller controls, exogenous butyrate supplementation (100 mM) rescued motility defects in intestinal polyQ44 worms (**Figs 2C and 5**). We further explored the ability of butyrate to suppress aggregation in other tissues by repeating the above experiment using the muscle polyQ model and assessed the effect of butyrate on bacteria-induced protein aggregation at

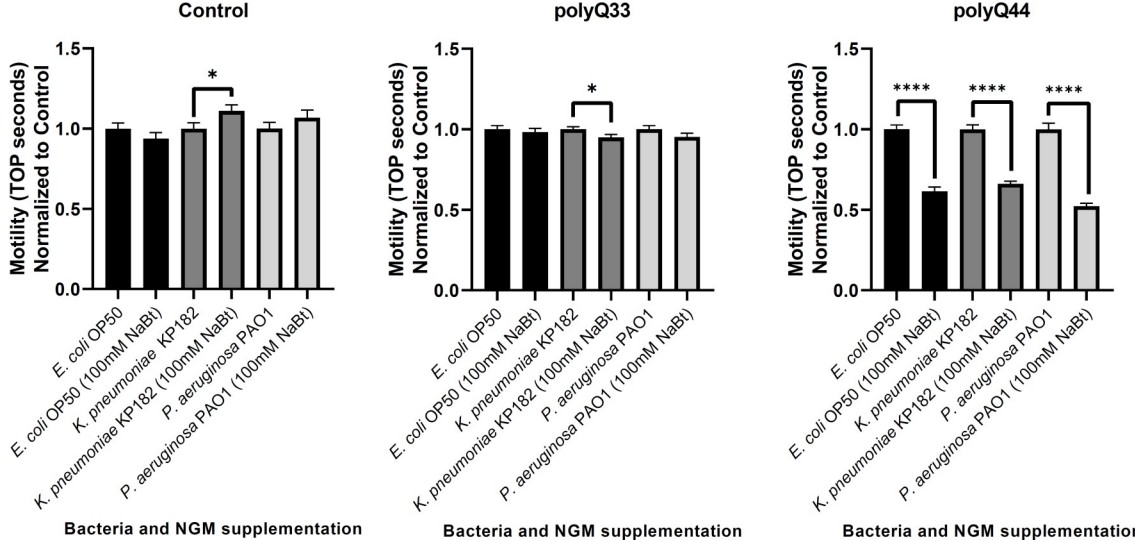

**Fig 5. The effect of butyrate on toxicity associated with bacteria-mediated intestinal polyQ aggregation.** Aggregation-dependent toxicity is assessed by measuring motility (TOP phenotype) in the presence of 100 mM butyrate. Data are represented as the average number of TOP seconds per worm normalized to the control (0 mM butyrate). Two roller strains, Control (AM446, no polyQ) and polyQ33 (no aggregates), are controls. Each bar is an average of three independent experiments with a total of 60 animals. Error bars represent SEM. Statistical significance between each pair was calculated using Student's t-test (*p<0.05, ****p<0.0001).

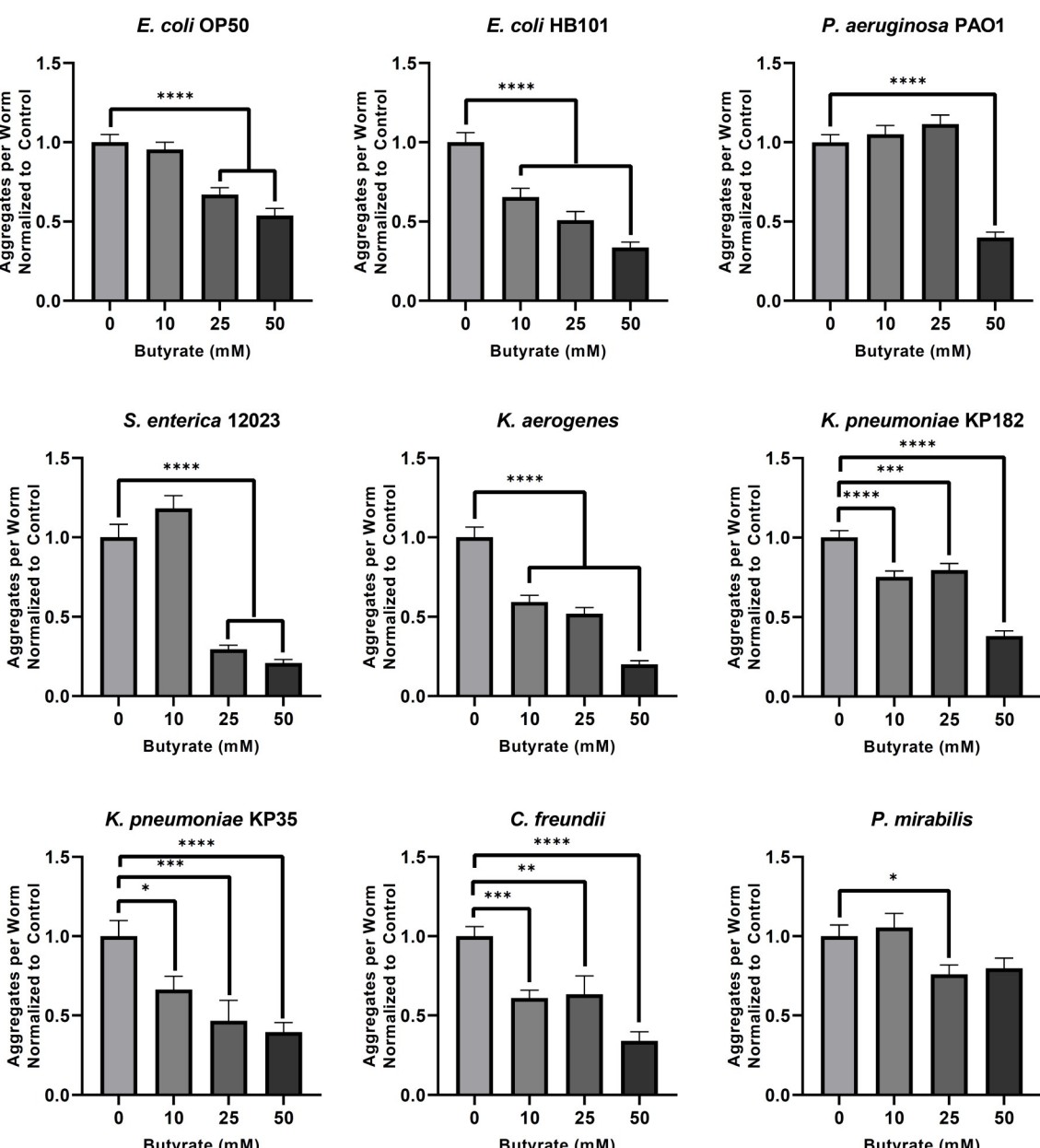

**Fig 6. Butyrate affects bacteria-mediated protein aggregation in the *C. elegans* muscle.** Data are represented as the average number of aggregates of muscle-specific polyQ35 (AM140) per worm normalized to the control (0 mM butyrate). Each bar is an average of three independent experiments with a total of 100 animals and the error bars represent SEM. Statistical significance was calculated using one-way ANOVA followed by multiple comparison Dunnett's post-hoc test (*$p<0.05$, **$p<0.01$, ***$p<0.001$, ****$p<0.0001$).

concentrations ranging from 0–50 mM. Unlike the intestinal polyQ44 model, worms expressing muscle polyQ35 did not experience a significant enhancement of aggregation at lower butyrate concentrations and exhibited a dose-response or near dose-response suppression of aggregation across all bacterial strains except for *P. mirabilis* and *P. aeruginosa* PAO1, wherein the only suppressive concentrations of butyrate were 25 and 50 mM, respectively (**Fig 6**). It is worth noting that the trends of butyrate-suppressed aggregation in worms expressing muscle polyQ parallel those seen in worms expressing intestinal polyQs (**Fig 4**). For example, animals

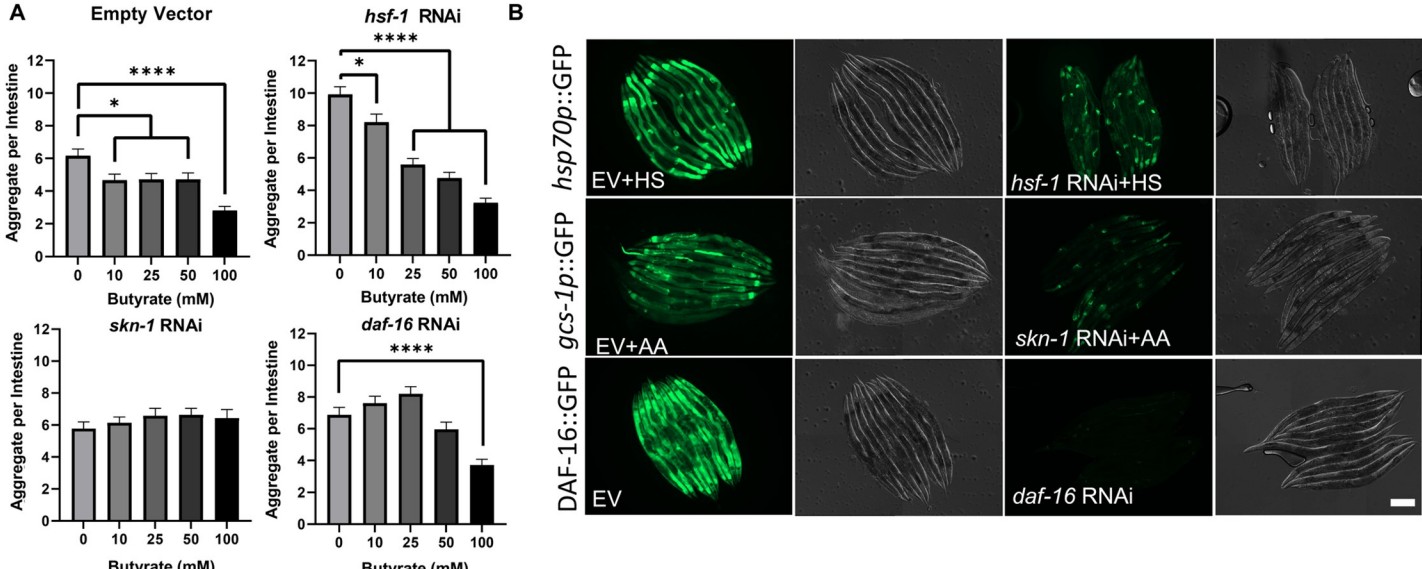

**Fig 7. SKN-1 and DAF-16 are involved in butyrate-mediated suppression of aggregation. A)** The effect of *hsf-1*, *skn-1*, and *daf-16* knockdown on intestine-specific polyQ44 aggregation in the presence of exogenous butyrate supplementation, compared to empty vector control (L4440). Data are represented as the average number of aggregates per *C. elegans* intestine. Each data point is an average of a minimum of three independent experiments with a total of at least 85 animals. Error bars represent SEM. Statistical significance was calculated using one-way ANOVA followed by multiple comparison Dunnett's post-hoc test ($^*$p<0.05, $^{****}$p<0.0001). **B)** Fluorescent and Nomarski images of worms expressing reporter constructs regulated by HSF-1, SKN-1, and DAF-16. Worms were fed *E. coli* expressing either the control empty vector (EV) shown in the left panels or specific RNAi shown in the right panels. To activate the reporters, animals expressing *hsp70p*::GFP and *gcs-1p*::GFP were either heat shocked (HS) or exposed to 5 mM acrylamide (AA), respectively. Functional knockdowns were confirmed by the attenuated GFP expression in animals fed RNAi versus EV. Scale bar = 200 μm.

cultured on plates containing 0–50 mM butyrate exhibited a significant dose-dependent decrease in intestinal and muscle aggregation when colonized by *E. coli* OP50 or *Klebsiella* spp. On the other hand, lower concentrations of butyrate did not significantly suppress aggregation in either tissue when animals were fed *P. aeruginosa* PAO1, *S. enterica* 12023, and *P. mirabilis*, suggesting that the beneficial effect of butyrate is, at least in part, facilitated by bacteria (**Figs 4** and **6**).

To gain a mechanistic understanding of how butyrate suppresses bacteria-mediated aggregation, we investigated the dependence of the observed response on three major evolutionary-conserved transcription factors, HSF-1, SKN-1/Nrf2, and DAF-16/FOXO. Each of these transcription factors was previously shown to provide protection against proteotoxicity [38–40]. Because butyrate suppressed aggregation upon colonization with multiple strains of bacteria, including *E. coli* OP50 (**Fig 4**), we investigated the dependence of butyrate-mediated suppression of aggregation on the abovementioned transcription factors by knocking-down each candidate using RNAi. *C. elegans* carrying intestinal polyQ44 that were fed *E. coli* HT115 (DE3) expressing empty vector control (L4440) RNAi, exhibited a significant suppression of aggregation in the presence of butyrate (10–100 mM) (**Fig 7A**). While butyrate-mediated suppression of aggregation was not affected when HSF-1 was knocked-down, the beneficial effect of butyrate was abolished upon downregulation of SKN-1/Nrf2 and DAF-16/FOXO transcription factors. As expected, HSF-1 knockdown enhanced aggregation in the absence of butyrate when compared to control animals (**Fig 7A**) [41]. To confirm that the knockdown is functional, we used fluorescent reporters to assess the expression of downstream target genes of HSF-1 (*hsp70p*::GFP also known as *C12C8.1p*::GFP) and SKN-1 (*gcs-1p*::GFP), and we used DAF-16

(DAF-16::GFP) fluorescent fusion (**Fig 7B**). The expression of all fluorescent reporters was attenuated upon knockdown of the respective target genes.

Together, these results indicate that butyrate can suppress bacteria-induced polyQ aggregation across tissues and rescues aggregate-dependent toxicity in the host. Such protective effect of butyrate seems to be mediated by SKN-1 and DAF-16 transcription factors.

## Colonization of the *C. elegans* intestine with *E. coli* engineered to overproduce butyrate suppresses protein aggregation across *C. elegans* tissues

SCFA-producing bacteria are common residents of the human gut microbiota. Many of these bacteria, such as those belonging to the Firmicutes phylum, are known to produce butyrate [42]. Butyrate has been demonstrated to provide numerous health benefits against a variety of ailments, and neurodegenerative diseases are no exception [32]. To determine the effect that butyrogenic bacteria have on organismal proteostasis, we used an *E. coli* strain, LW393 (*E. coli*$^{Bt}$), engineered to conditionally overproduce butyrate [43]. Colonizing *C. elegans* intestines with this strain allows us to model simplified physiological, yet controlled, conditions. The strain was derived from *E. coli* W (*E. coli*$^{WT}$), a non-pathogenic strain closely related to commensal bacteria by first creating a quadruple deletion mutant (*ldhA*, *frdABCD*, *ackA*, and *adhE*), *E. coli* BEM9 (*E. coli*$^{\Delta}$) strain, followed by overexpression of enzymes that channel intermediates into the pathway and which are essential for butyrate synthesis [43]. Therefore, we used *E. coli*$^{WT}$ and, in some instances, *E. coli*$^{\Delta}$, as controls. *E. coli*$^{Bt}$ synthesizes butyrate in the presence of 5- and 6-carbon sugar substrates to levels up to three times higher than any other available commercial strain [43]. We first used this strain to colonize the intestine of *C. elegans* expressing the intestinal polyQ44 reporter over a period of four days. We demonstrated that, on days three and four, *C. elegans* intestines are colonized by viable bacteria (**S11 Fig**). While glucose is a suitable substrate for butyrate synthesis, we found that it affects polyQ aggregation and thus cannot be utilized in our experiments (**S12 Fig**); however, L-arabinose is another butyrate synthesis substrate and does not affect aggregation. To confirm that L-arabinose from the NGM plates is available to intestinal bacteria, we colonized *C. elegans* intestines with bacteria that express an arabinose-inducible fluorescent reporter and cultured them in the absence of the sugar. We then transferred these animals onto plates containing *E. coli* OP50 and 3% L-arabinose and detected a strong induction of fluorescence, indicating that L-arabinose is accessible to the intestinal bacteria (**S13 Fig**). We supplemented NGM plates with 3% L-arabinose and quantified the number of aggregates after four days feeding polyQ-expressing animals the three control (*E. coli* OP50, *E. coli*$^{WT}$, and *E. coli*$^{\Delta}$) and the butyrate-producing (*E. coli*$^{Bt}$) strains. We did not detect any effect on polyQ aggregation in worms colonized with each of the control *E. coli* strains; however, supplementation of L-arabinose did result in significant suppression of polyQ aggregation in animals colonized by *E. coli*$^{Bt}$ (**Fig 8A**). To determine whether the aggregate-suppressing effect of *E. coli*$^{Bt}$ would also rescue polyQ-associated motility defects, we measured the TOP phenotype in worms expressing intestinal polyQ44. In agreement with the observed suppression of aggregation, supplementation of L-arabinose to animals colonized by *E. coli*$^{Bt}$ also improved motility, indicating that bacteria-derived butyrate can rescue aggregation and aggregate-associate toxicity (**Fig 8B**).

We further investigated whether the effect of endogenous butyrate produced by *E. coli*$^{Bt}$ strain was limited to the intestine or was ubiquitous across tissues, as seen when the animals were exposed to exogenous butyrate. To accomplish this, we colonized the intestines of animals expressing muscle and neuronal polyQ with *E. coli*$^{Bt}$. While supplementation of 3% L-arabinose into NGM plates did not affect intestinal polyQ aggregation in animals colonized by

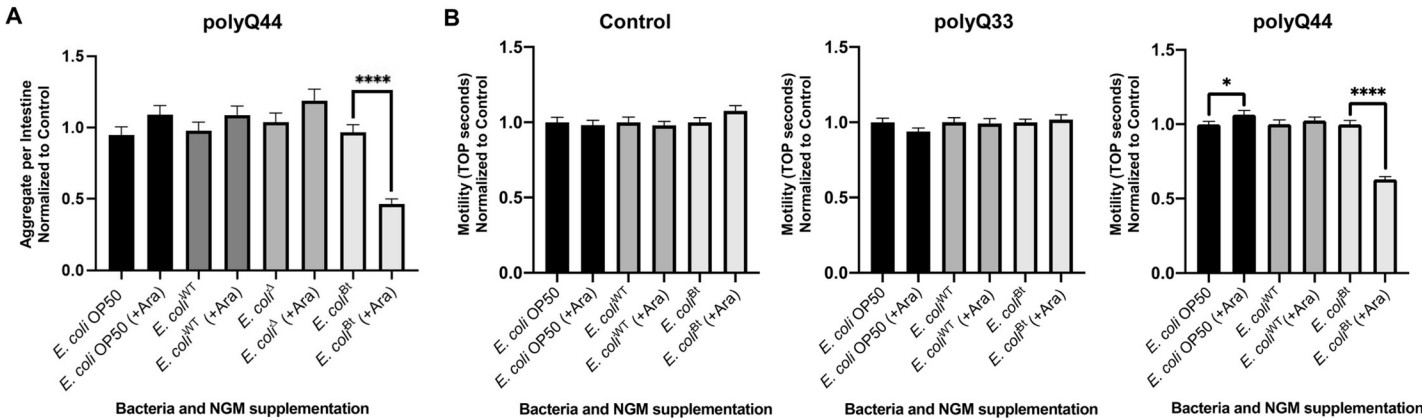

**Fig 8. Butyrate-producing *E. coli* suppresses aggregation and the associated toxicity.** Animals were fed four different strains of bacteria: non-butyrogenic controls (*E. coli* OP50, *E. coli*^WT, *E. coli*^A) and conditional butyrogenic *E. coli* (*E. coli*^Bt). **A)** The graphs represent the average number of intestinal polyQ44 aggregates per worm normalized to the control (no arabinose). Each bar is an average of three independent experiments with a total of 100 animals. **B)** Intestinal aggregate-dependent toxicity normalized to the control (no arabinose) assessed with the TOP phenotype. The left panel represents roller worms (Control), the middle panel represents polyQ33 worms, and the right panel represents worms expressing polyQ44. Each bar is an average of three independent experiments with a total of 60 animals. Error bars represent SEM. Statistical significance between each pair was calculated using Student's t-test (*p<0.05, ****p<0.0001).

control *E. coli* strains (**Fig 8A**), it did enhance aggregation in animals expressing muscle-specific polyQs (**Fig 9A**). Intriguingly, this enhancement of aggregation was significantly reduced in the presence of L-arabinose in animals colonized by *E. coli*^Bt, suggesting that endogenously produced butyrate suppresses protein aggregation in the muscle tissue as well (**Fig 9A**). We then assessed the effects of this strain on aggregate-dependent toxicity in this model utilizing the body bend assay. Our results show that worms expressing muscle-specific polyQ have 2.5x more body bends when *E. coli*^Bt is endogenously producing butyrate, compared to worms colonized with non-butyrogenic bacteria or butyrogenic bacteria in the absence of the sugar substrate (**Fig 9B**). The observed beneficial effect of endogenous butyrate is dependent on the presence of polyQ since endogenous butyrate did not increase the number of body bends in control wild-type worms (**Fig 9C**). In fact, the presence of endogenous butyrate was slight, but significantly detrimental to wild-type *C. elegans*. These results support the beneficial effect of butyrate in the suppression of aggregation and the associated proteotoxicity in the muscle tissues.

We expanded our investigation to animals carrying a neuronal polyQ40 reporter by measuring toxicity using the TOP phenotypic readout. We employed this method to assess the effect of endogenous butyrate on aggregate-dependent toxicity. The availability of L-arabinose to control *E. coli* strains (*E. coli* OP50 and *E. coli*^WT) did not improve motility. Still, the butyrate produced by *E. coli*^Bt significantly enhanced motility in animals carrying the neuronal polyQ reporter (**Fig 9D**), but not in N2 control worms (**Fig 9E**). While further investigation is needed to elucidate the basis of this response, the result could be key to understanding the mechanism(s) by which butyrogenic bacteria affect host proteostasis.

## Co-colonization of the *C. elegans* intestine with butyrogenic bacteria suppresses bacteria-induced aggregation

The human gut microbiota is a polymicrobial community and the effect of the interactions between individual strains and the host is hindered by the complexity and wealth of its composition. We co-colonized the *C. elegans* gut with pathogenic and butyrogenic bacteria as a strategy to begin deconvoluting complex polymicrobial interactions within the human

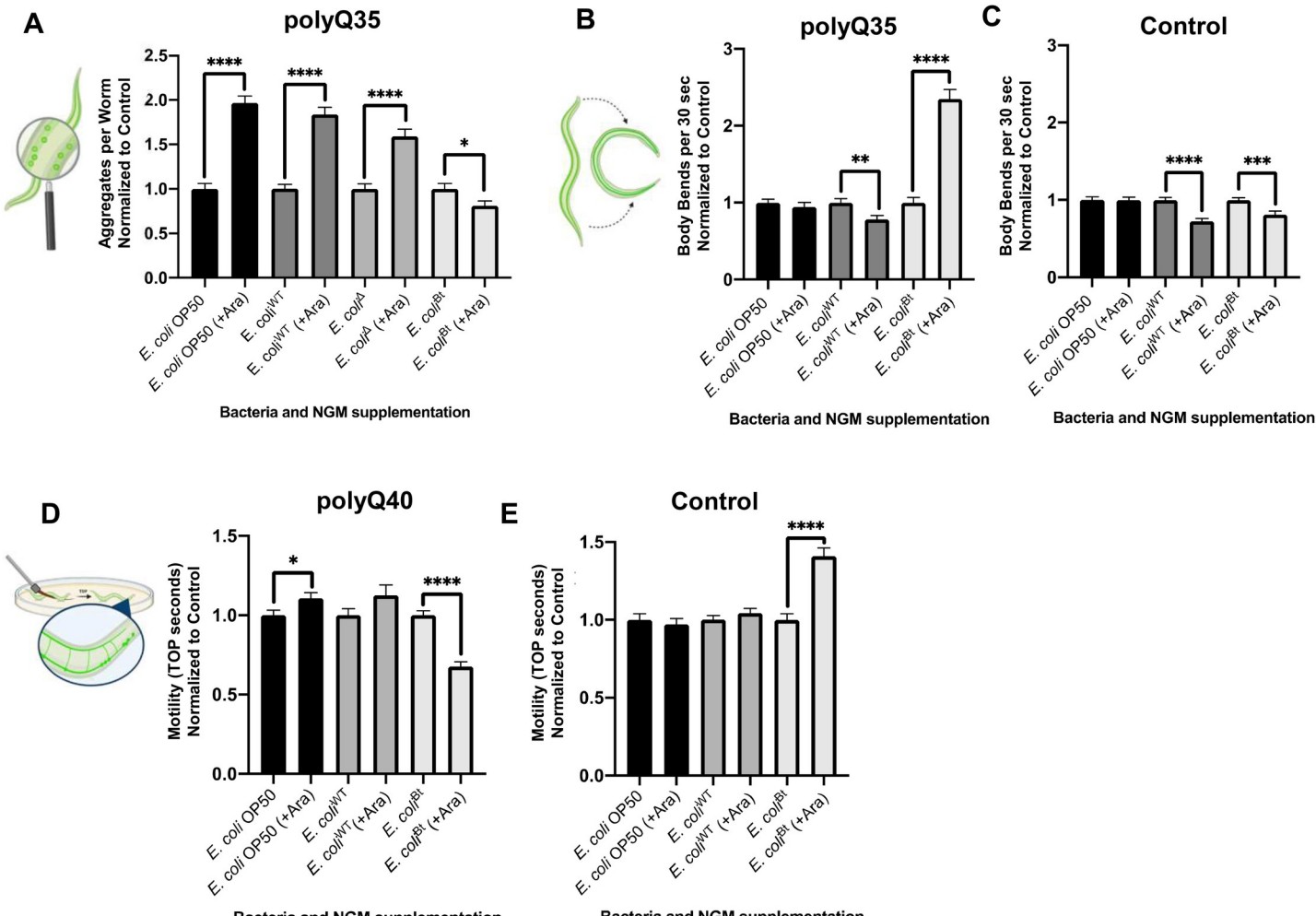

**Fig 9. Colonization of the *C. elegans* intestine with butyrogenic *E. coli* suppresses aggregation and associated toxicity in distal tissues. A)** The effect of butyrogenic bacteria (*E. coli*[Bt]) on the aggregation profile of worms expressing muscle-specific polyQ35. Data are represented as the average number of aggregates of muscle-specific polyQ35 (AM140) per worm normalized to the control (no arabinose). Animals were fed four different strains of bacteria: non-butyrogenic controls (*E. coli* OP50, *E. coli*[WT], *E. coli*[Δ]) and conditional butyrogenic *E. coli* (*E. coli*[Bt]). **B)** The effect of butyrogenic bacteria on the motility of animals expressing muscle-specific polyQ35 and **C)** control N2 worms. Data are represented as the average number of body bends per worm normalized to control (no arabinose). Each bar is an average of three independent experiments with a total of 45 animals. **D)** The effect of butyrogenic bacteria on the motility of animals expressing neuronal polyQ40 and on **E)** control N2 worms. Data are represented as the average TOP seconds per worm normalized to control animals (no arabinose). Error bars represent SEM. Statistical significance between each pair was calculated using Student's t-test (*p<0.05, **p<0.01, ***p<0.0005 ****p<0.0001).

microbiome and their effect on the host. We fed worms cultures of *E. coli*[Bt] with either *E. coli* OP50 as a control or *P. aeruginosa* PAO1, which showed the most robust enhancement of aggregation. In the presence of L-arabinose, we found a significant decrease in aggregation of intestinal polyQs in co-cultures with both strains (**Fig 10A**). *E. coli*[Bt] suppressed *E. coli* OP50- and *P. aeruginosa* PAO1-mediated aggregation by approximately 4- and 2-fold, respectively. These results indicate that endogenous butyrate synthesized by butyrogenic bacteria can enhance host proteostasis and suppress protein misfolding. Furthermore, our experiments demonstrate the potential benefits of butyrate and butyrogenic bacteria in the suppression of bacteria-induced proteotoxicity associated with metastable proteins and also emphasize the importance of having a balance between commensal butyrogenic and enteropathogenic microbes (**Fig 10B**).

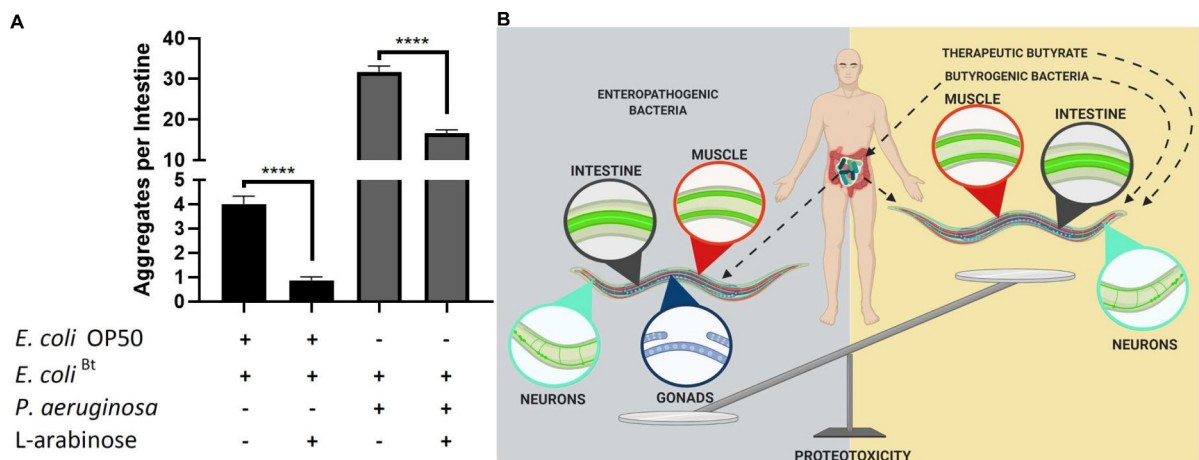

**Fig 10. Co-colonization of *E. coli*[Bt] suppresses *P. aeruginosa*-mediated enhancement of polyQ aggregation. A)** The effect of butyrogenic bacteria (*E. coli*[Bt]) on the aggregation of intestinal polyQ44 (AM738) in worms co-colonized with *E. coli* OP50 or *P. aeruginosa* in the presence and absence of 3% L-arabinose. Data are represented as the average number of aggregates per intestine. Each bar is an average of three independent experiments with a total of 100 animals. Error bars represent SEM. Statistical significance between each pair was calculated using Student's t-test (****p<0.0001). **B)** Model figure. Enteropathogenic bacteria disrupt proteostasis across *C. elegans* tissues, including intestine, muscle, neurons, and gonads. The bacteria-mediated proteotoxicity is alleviated by butyrate and butyrogenic bacteria.

## Discussion

We used a simple metazoan model, *C. elegans*, to study the effect of bacterial gut colonization on organismal proteostasis. We found that the colonization of the *C. elegans* intestine with enteric bacterial pathogens disrupts organismal proteostasis and enhances proteotoxicity associated with polyQ. Importantly, our results demonstrate that bacteria alone do not cause major toxicity in control animals that do not express polyQ, which would hinder the interpretation of our data (**Figs 2B, 2C**, **3C**, and **3F**). It is not surprising that the detrimental effect of bacteria on polyQ aggregation is observed in the most proximal tissue, the intestine, as similar results were previously observed [23]. However, to our knowledge, this is the first report of bacteria having detrimental, tissue non-autonomous effects on protein folding. Recently, *Bacillus subtilis*, a probiotic strain that is also part of the human commensal microbiota, was shown to inhibit α-synuclein aggregation in *C. elegans* muscle [44]. The fact that certain bacteria disrupt host proteostasis while others can enhance it emphasizes the importance of microbes in the pathogenesis of PCDs. Previous research suggests that increased presence of bacterial species such as *K. pneumoniae* and *P. aeruginosa* in the human gut is associated with an increased prevalence of PCDs [45,46]. Interestingly, among all of the strains that we tested, *Klebsiella* spp. and *P. aeruginosa* were the most potent inducers of polyQ aggregation–this was the case even for a commensal strain, *K. oxytoca* (**Fig 1B and 1C**). Both of these species are ubiquitous in the environment, and multidrug-resistant strains of these bacteria are often associated with nosocomial and opportunistic infections [47]. Due to their rapidly increasing resistance to antibiotics, they are classified by the Centers for Disease Control and Prevention as serious threats [48]. While these strains are often associated with infections, their enteric presence is also evident in healthy individuals and may possibly support their contribution to neurodegenerative diseases [49,50]. In fact, both of these microbes have recently been associated with neurodegenerative diseases in humans [45,46], a correlation that is in agreement with our results. The increasing prevalence of antibiotic resistance among *Klebsiella* spp. and *P. aeruginosa* strains may specifically enrich for their growth within the human microbiome upon an antibiotic treatment while decreasing the abundance of beneficial bacteria. In agreement with

this hypothesis, population-based studies of patients with Parkinson's and ALS revealed that a history of antibiotic treatment is associated with an elevated risk for these diseases [7,8].

Another notable strain that induced aggregation by more than three-fold is *P. mirabilis*. This gram-negative bacterium is part of human commensal microbiota but can also become an opportunistic pathogen, most often leading to urinary tract infections, and eventually, bacteremia [51]. Interestingly, in one study, blood cultures of PD septic patients had a five-fold higher prevalence of *P. mirabilis* compared to non-PD septic patients [52]. Recently, it was demonstrated in a mouse model that *P. mirabilis* contributes to PD pathogenesis, including the induction of α-synuclein in the mouse brain [53]. Furthermore, colonization of the gut with *P. mirabilis* and *Enterobacteriaceae*, in particular, *K. pneumoniae*, was shown to be accelerated by antibiotic treatment, leading to detrimental dysbiosis in mouse microbiota [54]. On the other side of the aggregation spectrum, the non-pathogenic *E. coli* HB101 did not significantly enhance aggregation in our *C. elegans* polyQ model. The *E. coli* HB101 strain is also a common laboratory food for *C. elegans*; hence, we did not expect any enhancement with these bacteria. However, both *P. disiens* HM-1171 and *P. corporis* HM-1294 suppressed aggregation relative to *E. coli* OP50 (**Fig 1C**). The mechanism by which these *Prevotella* spp. suppress aggregation does not depend on butyrate as both strains were deficient in its production [55,56]. Interestingly, *Prevotella* was the only bacterial genus whose abundance significantly decreased in patients suffering from constipation, a condition that is a hallmark in Parkinson's disease and is associated with up to 80% of cases [57–59]. Further evidence that these two species could be beneficial to host proteostasis comes from a recent study that demonstrated a negative correlation between the abundance of *Prevotella* in the gut of Parkinson's disease patients and the severity of symptoms [60]. A negative correlation between the abundance of *Prevotella* and pathogenesis of Alzheimer's disease mouse model has also been observed [61]. Collectively, these results suggest that the members of the *Prevotella* genus could play a more direct role in enhancing host proteostasis, and that the mechanisms by which bacteria affect host proteostasis in our *C. elegans* model may be conserved in higher organisms.

Although *P. aeruginosa* was previously found to induce neurodegeneration in *C. elegans* [62], the results of decreased polyQ-dependent motility of animals colonized with *P. aeruginosa* and *K. pneumoniae* were unexpected and suggest that either bacteria, or host responses to these bacteria, affect the function of distal tissues (i.e., muscle or neurons), but only in animals that express metastable aggregation-prone proteins such as polyQs (**Figs 2B, 2C, and 3A–3G**). Recent studies show that colonization of the *C. elegans* intestine with pathogenic bacteria affects bacterial avoidance behavior [63] and consequently increases lifespan [64]. While it is possible that feeding behavior might have been affected in our experiments, it is unlikely that avoidance would contribute to an increase in polyQ aggregation and the associated motility defects. In fact, if the animals in our experiments experienced food avoidance behavior, we would expect to see enhanced proteostasis, as dietary restriction was previously shown to suppress polyQ aggregation [65].

It is known that bacterial colonization of the *C. elegans* intestine increases the production of reactive oxygen species (ROS), which in turn enhances aggregation in that tissue [23]. Oxidative stress is known to affect overall animal physiology, including motility [66]. Since our results show the bacteria-mediated motility defects only in the presence of aggregation-prone polyQ44, but not in non-aggregating polyQ33 or non-polyQ controls (**Fig 2C**), it is possible that ROS generated by the worm in response to bacterial colonization enhanced polyQ aggregation, which further affected the function of other tissues involved in motility. Such a mechanism likely contributes to the observed bacteria-mediated disruption of host proteostasis which is supported by our results demonstrating that butyrate rescued polyQ aggregation in a DAF-16 and SKN-1-dependent manner (**Fig 7A**). However, ROS is likely not the sole

contributor since our results also indicate that bacterial aggregates affect host proteostasis and consequently increase aggregation of host polyQ (**Fig 1E–1G**).

Tissue non-autonomous regulation has been noted in *C. elegans* under various conditions [27,67]. Our results demonstrating that bacterial colonization of the *C. elegans* intestine affects proteostasis across tissues are intriguing, as they suggest that bacteria impose tissue non-autonomous effects on the host. In human and mouse studies, the effect of bacterial gut colonization on host proteostasis has been primarily attributed to non-specific mechanisms, such as systemic inflammation, which broadly affects many tissues [68]; however, *C. elegans* lacks an inflammatory response, indicating that other and possibly more direct mechanisms may be involved. Bacterial amyloids have also been shown to specifically contribute to pathogenesis by affecting metastable aggregation-prone proteins [69]. The introduction of metastable proteins is known to affect the stability and folding of other proteins within the proteome [70,71]. As such, it is likely that bacteria may be secreting factors that translocate from the intestine across other tissues and affect proteostasis. A cross-seeding between disease-associated proteins (i.e., Aβ) with bacterial amyloids, including those from *E. coli* and *P. aeruginosa*, was previously observed [72,73]. Also, an enhancement of alpha-synuclein aggregation mediated by bacterial curli was shown in mouse and *C. elegans* Parkinson's disease models [74]. While our results indicate that *E. coli* curli fimbriae were not responsible for the enhancement of polyQ aggregation, *E. coli* aggregates did play a contributing factor (**Fig 1G**). It is possible that alpha-synuclein and polyQ are affected differently by bacterial amyloids. In addition to regulating the *csg* operon, CsgD also functions as a global transcriptional regulator and its target gene(s), other than *csgA*, could affect host proteostasis [75]. It is intriguing to speculate that the transmission of aggregation-prone proteins between tissues [67], may underlie how bacterial colonization of parental intestines affects protein aggregation in the progeny (**Fig 3I**).

A symbiosis between commensal and pathogenic bacteria can be disrupted by many factors, including infections, diet, and antibiotics [76]. While dysbiosis of the human gut microbiota has been associated with the pathogenesis of PCDs, the mechanisms that contribute to the disruption of host proteostasis and disease progression remain poorly understood. Our results demonstrate that butyrate, a common metabolite produced by commensal microbiota, can suppress aggregation and the associated toxicity when supplied exogenously (**Figs 4 and 6**) or produced by intestinal bacteria (**Figs 8 and 9**). We also showed that butyrate did not simply inhibit intestinal colonization (**S9 Fig**). In agreement with our results, the protective effects of butyrate have been previously demonstrated in a mouse model and in humans [21,32]. Our data indicate that the benefits of butyrate depend on SKN-1/Nrf2 and DAF-16/FOXO transcription factors, suggesting that butyrate may inhibit bacteria-induced aggregation by activating protective stress responses, in particular, oxidative stress responses. Indeed, bacteria are known to trigger oxidative stress, which can contribute to polyQ aggregation [23]. It is possible that various bacteria introduce different levels of oxidative stress and the strongest contributors affect host proteostasis, consequently leading to misfolding and aggregation of metastable proteins present within the proteome. In fact, oxidative stress is one of the major contributors to PCDs [77]. Recently, it was found that butyrate and β-hydroxybutyrate provide Nrf2- and FOXO-dependent protection against oxidative stress, respectively [78,79]. In addition, both SKN-1 and DAF-16 were shown to be involved in β-hydroxybutyrate-mediated lifespan extension [80]. These results, along with our data, collectively suggest that butyrate activates SKN-1/Nrf2 and DAF-16/FOXO, which attenuate bacteria-induced oxidative stress. The effect of butyrate on polyQ aggregation also seems to depend on the type of pathogenic bacteria present, suggesting that butyrate may contribute to host proteostasis by acting on bacteria in addition to the host. These results are consistent throughout the intestinal and muscle-specific polyQ models. Additionally, the co-colonization experiment further supports that the extent

of the beneficial effect of butyrate depends on the bacteria present (**Fig 10A**). The fact that bacteria are viable in the presence of butyrate at concentrations that suppress aggregation (**S6 and S7 Figs**) and that dead bacteria exhibit a distinct effect on aggregation (**S8 Fig**) suggest that microbial processes which affect the host may be targeted by butyrate. Whether butyrate affects specific bacterial signals that contribute to PCDs remains to be determined and further studies are underway to identify bacterial factors that enhance aggregation of host metastable proteins. Whatever the mechanism may be, our results suggest a therapeutic potential of butyrate in the prevention or treatment of PCDs.

High metabolic rates of bacteria can lead to depletion of available oxygen and introduce a hypoxic environment on the plates. Low oxygen levels are known to disrupt cellular processes leading to increased polyQ aggregation [81]. One of the possible explanations regarding how butyrate may inhibit bacteria-mediated polyQ aggregation is by suppressing bacterial metabolism, which restores the normoxic environment. However, if this were true, we would expect to have seen the restoration of a normoxic environment when we fed worms heat-killed bacteria, which would result in polyQ aggregation similar to 100 mM butyrate (**S8 Fig**). Therefore, it is likely that butyrate could have some additional effects on bacteria in addition to the host. In fact, it is known that butyrate can affect bacterial pathogenicity [34,82,83]. Further research is underway to determine what changes butyrate might exert on bacteria and how that influences the host.

## Methods

### Bacterial strains

Detailed information about bacterial strains used and generated in this study is provided in Table 1.

### *Caenorhabditis elegans* maintenance and strains

Nematodes were maintained according to previously established protocols [89]. All experiments were performed at room temperature (RT, ~23 ºC) unless described otherwise. All *C. elegans* strains used in this study can be found in Table 1.

### Bacterial culture conditions

Culture conditions for bacteria used in endogenous and exogenous butyrate treatment can be found in "Endogenous Butyrate Production" and "Exogenous Butyrate Treatment", respectively. DC228 was inoculated in the presence of 2 µg/mL gentamicin. Strains were plated onto Nematode Growth Media (NGM) +/- 3% arabinose and were left to dry overnight at RT prior to having worms plated on them. Curli-mutant bacteria and wild-type control (MG1655) were plated on NGM and kept at RT for 2 days prior to having worms plated on them. *K. oxytoca* (HM-624), *Prevotella disiens* (HM-1171), *Prevotella corporis* (HM-1294), and their *E. coli* OP50 control were grown in Reinforced Clostridial Broth (RCB). Unless otherwise indicated, all other bacteria were grown overnight in a 37˚C incubator shaking at 220 revolutions per minute (RPM) in Lennox Luria Broth (LB), were seeded on NGM, left to dry overnight, and were kept at 4˚C for another day prior to having worms plated on them.

### Transduction and verification of curli (-) mutants

The bacterial strains used in this study are listed in Table 1. All solutions were prepared using ultrapure water, 18.2 MΩ/cm (Barnstead). Bacterial strains were grown in LB (5 g yeast extract, 10 g Bacto tryptone, and 10 g NaCl per liter of water), pH 7.4, at 37˚C, with shaking (250

**Table 1. Reagents and resources used in this study and their associated sources and identifiers.**

| REAGENT or RESOURCE | SOURCE | IDENTIFIER |
|---|---|---|
| Bacterial Strains | | |
| *Escherichia coli* OP50 | Caenorhabditis Genetics Center | WB OP50; RRID: WB-STRAIN: OP50; NCBI TaxID: 637912; DC199 |
| *Escherichia coli* HB101 | Caenorhabditis Genetics Center | WormBase ID: WBStrain00041075; DC210 |
| *Escherichia coli* E22 | Shuman Lab, (University of Chicago) | E22, DC66 |
| *Escherichia coli* E2348/69 | Shuman Lab (University of Chicago) | E2348/69, DC67 |
| *Klebsiella aerogenes* | Shuman Lab (University of Chicago) | NCBI TaxID: 548; DC134; MG917 |
| *Klebsiella pneumoniae* KP35 | Prince Lab (Columbia University) | NCBI TaxID: 1460422; KP35/ST258; DC86 |
| *Klebsiella pneumoniae* KP182 | Shuman Lab (University of Chicago) | NCBI TaxID: 1352932; DC4 |
| *Klebsiella pneumoniae* KCI-1 | University of Chicago Clinical Microbiology | F50024/KCI-1, DC94 |
| *Klebsiella pneumoniae* KCI-2 | University of Chicago Clinical Microbiology | W70979/KCI-2, DC95 |
| *Klebsiella pneumoniae* KCI-3 | University of Chicago Clinical Microbiology | W71530/KCI-3, DC96 |
| *Klebsiella pneumoniae* KCI-4 | University of Chicago Clinical Microbiology | T32014/KCI-4, DC97 |
| *Klebsiella pneumoniae* KCI-5 | University of Chicago Clinical Microbiology | F53563/KCI-5, DC98 |
| *Proteus mirabilis* Hauser 1885 | ATCC | ATCC 35659 NCBI TaxID: 584; DC225 |
| *Citrobacter freundii* | NorthShore Research Institute | ACI-5, DC153 |
| *Shigella sonnei* Levine 1920 | ATCC | ATCC 9290, DC227 |
| *Salmonella* Typhimurium 12023 | ATCC | ATCC 14028; DC216 |
| *Pseudomonas aeruginosa* PAO1 | Shuman Lab (University of Chicago) | PAO1; DC3 |
| *Acinetobacter baylyi* ADP1 | ATCC | ATCC 33305; ADP1; DC167 |
| *Acinetobacter baumannii* AB5075 | Shuman Lab (University of Chicago) | AB5075; DC1 |
| *Acinetobacter baumannii* 17978 | Shuman Lab (University of Chicago) | 17978; DC2 |
| *Escherichia coli* W | Shanmugam Lab (University of Florida) | ATCC9637, *E. coli*[W], DC222 |
| *Escherichia coli* LW393 | Shanmugam Lab (University of Florida) | *E. coli*[Bt], DC208 |
| *Escherichia coli* BEM9 | Shanmugam Lab (University of Florida) | *E. coli*[A], DC223 |
| *Escherichia coli* DC228 (LW393-pMJG125.sfGFP) | This study | DC228 |
| *Klebsiella oxytoca* | BEI Resources | HM-624, MIT 10–5243 |
| *Prevotella disiens* | BEI Resources | HM-1171, DNF00882 |
| *Prevotella corporis* | BEI Resources | HM-1294, MJR7716 |
| *Escherichia coli* HT115 | Ahringer RNAi Library [84] | *skn-1* (T19E7.2) RNAi |
| *Escherichia coli* HT115 | Ahringer RNAi Library [84] | *daf-16* (R13H8.1) RNAi |
| *Escherichia coli* HT115 | Ahringer RNAi Library [84] | *hsf-1* (Y53C10A.12) RNAi |
| *Escherichia coli* HT115 | Ahringer RNAi Library [84] | Empty vector, L4440 |
| *Escherichia coli* MG1655 | CGSC (no. 6300) | *E. coli* MG1655, F⁻ λ⁻ *rph-1* |
| *Escherichia coli* BW25113 Δ*csgA* | [85] | BW25113 Δ*csgA::kan* |
| *Escherichia coli* BW25113 Δ*csgD* | [85] | BW25113 Δ*csgD::kan* |
| *Escherichia coli* MG1655 Δ*csgD* | This study | *E. coli* MG1655 Δ*csgD::kan* |
| *Escherichia coli* MG1655 Δ*csgA* | This study | *E. coli* MG1655 Δ*csgA::kan* |
| *C. elegans* Strains | | |
| *C. elegans* N2, Bristol | Caenorhabditis Genetics Center | N2 |
| *C. elegans*: AM738: rmIs297[vha-6p::q44::yfp; rol-6(su1006)] | Morimoto Lab (Northwestern University) | AM738, Q44::YFP |

(*Continued*)

**Table 1.** (Continued)

| REAGENT or RESOURCE | SOURCE | IDENTIFIER |
|---|---|---|
| *C. elegans*: AM712 *rm1s281[vha-6p::q33::yfp; rol-6(su1006)]* | Morimoto Lab (Northwestern University) | AM712, Q33::YFP |
| *C. elegans*: AM446 *rmIs223[C12C8.1p::gfp;rol-6(su1006)]* | Morimoto Lab (Northwestern University) | AM446, *hsp70p*::GFP |
| *C. elegans*: AM140 *rmIs132[unc-54p::q35::yfp]* | Morimoto Lab (Northwestern University) | AM140, Q35::YFP |
| *C. elegans* AM141 *rmls133[unc-54p::q40::yfp]* | Morimoto Lab (Northwestern University) | AM141, Q40::YFP |
| *C. elegans*: AM101 *rmIs110 [F25B3.3p::q40::yfp]* | Morimoto Lab (Northwestern University) | AM101, Q40::YFP |
| *C. elegans*: DA597 *phm-2(ad597)* | Caenorhabditis Genetics Center | DA597 |
| *C. elegans*: LD1171 *ldIs3[gcs-1p::gfp+rol-6(su1006)]* | Caenorhabditis Genetics Center | LD1171, *gcs-1p*::GFP |
| *C. elegans* TJ356 *zIs356[daf-16p::daf-16a/b::gfp + rol-6(su1006)]*, | Caenorhabditis Genetics Center | TJ356, DAF-16::GFP |
| Chemicals & Commercial Assays | | |
| Levamisole | Fisher Scientific | Cat#ICN15522805 |
| Cholesterol | Fisher Scientific | Cat#ICN10138201 |
| Sodium butyrate | Fisher Scientific | Cat#A11079-22 |
| Thermo Scientific—AnaeroPack | Fisher Scientific | Cat#23-246-376 |
| Triton X-100, Molecular Biology Grade | Promega | Cat#H5141 |
| ProSignal Blotting Film | Prometheus | Cat#30-810L |
| Powdered nonfat milk | Research Products International | M17200-1000 |
| Tween-20 | Fisher BioReagents | Cat#C58H114O26 |
| Trans-Blot Turbo Midi-size Transfer Stacks | BioRad | Cat#1704273 |
| Trans-Blot Turbo Midi-size PDVF Membrane | BioRad | Cat#10026933 |
| Trans-Blot Turbo 5x transfer buffer | BioRad | Cat#10026938 |
| Criterion XT Precast Gel | BioRad | Cat#3450124 |
| XT 4x Sample Buffer | BioRad | Cat#1610791 |
| 20X Reducing Agent | BioRad | Cat#1610792 |
| XT MOPS | BioRad | Cat#1610788 |
| Clarity Western ECL | BioRad | |
| Isopropyl β-D-1-thiogalactopyranoside (IPTG) | GoldBio | Cat #12481C5 |
| Congo Red | Acros Organics | Cat#22962–0250 |
| Brilliant Blue G-250 | Fisher Biotech | CAS#6104-58-1 |
| Luria broth (Lennox) | Apex | Cat#11–125 |
| ProteoStat Protein aggregation assay | Enzo | Product #ENZ-51023-KP002 |
| Antibodies | | |
| Goat anti-mouse HRP secondary antibody | Thermo Scientific | Prod#31430 |
| Living Colors JL-8 primary monoclonal antibody | Takara Bio | Cat#632381 |
| Plasmids | | |
| pMJG125.sfGFP | This study | N/A |
| pBSK-sfGFP | Thomas Bernhardt (Harvard Medical School) | N/A |
| pKD46 | [86] | N/A |
| pXDC18.mCherry | [87] | N/A |
| P1*vir* | [88] | N/A |
| Software and Algorithms | | |
| GraphPad Prism v8.4.3 | GraphPad Software, Inc | https://www.graphpad.com |
| BioRender | BioRender | www.biorender.com |

(*Continued*)

**Table 1.** (Continued)

| REAGENT or RESOURCE | SOURCE | IDENTIFIER |
|---|---|---|
| Oligonucleotides | | |
| *ATAATAATTAAT*GTTATTGTCTCATGAGCGGATACA | IDT | MJG368 |
| ATCCATATGTTATAACCTCCTTAGAGC | IDT | MJG369 |
| ATTTATCATATGTCTAAAGGTGAAGAACTGTTCACCG | IDT | MJG1004 |
| TAAAATTCTAGATTATTTGTAGAGCTCATCCATGCCGTG | IDT | MJG1005 |
| CAGTATTTCGCAAGGTGCTTATG | IDT | *csgA conf* fwd |
| CCCTTGCTGGGTCGTATTAAA | IDT | *csgA conf* rev |
| GCAACATCTGTCAGTACTTCTGG | IDT | *csgD conf* fwd |
| CAGTATGGTCAGTTAGCAATCCC | IDT | *csgD conf* rev |

RPM), unless otherwise indicated. Transduction with P1vir was done to introduce gene deletions from *E. coli* donor strains from the Keio library [85]. Gene deletions were confirmed using PCR amplification of the target gene sequence using the primers listed in Table 1. The PCR products were visualized on an ethidium bromide-stained agarose gel.

### Congo red assay

Curli production of each strain was determined by Congo Red plate assay. Congo Red plates were made using LB medium without salt (5 g yeast extract and 10 g Bacto tryptone per liter of ultrapure water), to which 50 µg/mL filter sterilized Congo Red and 1 µg/mL filter sterilized Brilliant Blue G250 were added after autoclaving. Overnight cultures were pelleted and washed once in LB without salt. Cells were then resuspended in LB without salt and cell density (measured by $OD_{600}$) was normalized across all samples. Twenty-five microliters of each strain was spotted in a separate quadrant of Congo Red agar plate, which was then incubated at 26˚C for 48 h. Plates were imaged after 48 h incubation at 4˚C.

### ProteoStat assay

All bacteria were grown in LB in a 37˚C incubator shaking at 220 RPM overnight. Bacteria were diluted to $OD_{600}$ of 0.5 and seeded in the center of a 96-well plate. Outer wells were filled with water and the plate was sealed with parafilm. For consistency, each bacterial strain was tested in a separate plate. Plates were incubated for ~44 h at 26˚C. Following the incubation, LB was removed, and wells were washed twice with $ddH_2O$. Adhered biofilm was resuspended by pipetting with 100 µL 1x ProteoStat Assay Buffer. Two microliters of 1x ProteoStat Detection Reagent was dispensed into wells in a black, clear-bottom 96-well plate and 98 µL of resuspended biofilm was added and mixed. The plate was incubated for 15 minutes (min) at RT away from light. Following the incubation, fluorescence was read at excitation wavelength 550 nm and emission wavelength 600 nm, gain 100, in a Tecan Infinite 200 Pro plate reader. Assays were performed in triplicates.

### Plasmid construction

pXDC18.mCherry was digested with EcoRI and AseI to remove *lacI*[q] and P*tac*. A cassette harboring the *araC* gene and P*araBAD* promoter from plasmid pKD46 was amplified with primers MJG368 and MJG369, digested with EcoRI & AseI and cloned into the pXDC18.mCherry backbone to yield plasmid pMJG125. The sfGFP cassette from plasmid pBSK-sfGFP was amplified with oligonucleotide primers MJG1004 and MJG1005. The resulting PCR product

was digested with NdeI and XbaI and subsequently cloned into pMJG125 to yield pMJG125. sfGFP.

## Generation of bacterial strains

The arabinose-inducible *E. coli* reporter strain (DC228) was generated by the electroporation of arabinose-inducible fluorescent reporter encoded on plasmid pMJG125.sfGFP into *E. coli* LW393 using the BioRad Gene Pulser Xcell electroporator according to the manufacturer's instructions.

## Exogenous butyrate treatment

All bacteria were grown aerobically in LB and seeded on NGM plates supplemented with indicated concentrations of sodium butyrate or controls lacking the compound. To ensure uniform coverage of the bacterial lawn, cultures were spread evenly across the plate. Plates were left to dry overnight at RT and were placed at 4˚C one day prior to plating worms. Control experiments compared the aggregation profiles of worms fed heat-killed bacteria versus bacteria cultured in the presence of 100 mM sodium butyrate. Bacterial cultures were collected in 15 mL conical tubes and were killed by heating for 60 min at 70˚C water bath. Bacterial viability was assessed by inoculating 10% of the culture in LB followed by overnight growth at 37˚C shaking at 220 RPM. No growth confirmed bacteria to be dead. To bypass developmental delay that could result from feeding worms dead bacteria, animals were age synchronized, and cultured on plain NGM seeded with *E. coli* OP50 at 20˚C for 48 h. After 48 h, animals were washed 6x with 5 mL M9 and transferred to NGM plates containing either heat-killed bacteria or 100 mM sodium butyrate with live bacteria. Worms were incubated at RT for three additional days and aggregates were quantified as described in "Aggregate Quantification".

## Endogenous butyrate production

Overnight cultures of *E. coli*$^{Bt}$ and control strains, *E. coli*$^{WT}$ and *E. coli* OP50, were grown anaerobically. *E. coli* OP50 was grown anaerobically only in experiments where it was used as a control for *E. coli*$^{Bt}$. *E. coli*$^{\Delta}$ was grown aerobically, because this strain is not capable of anaerobic growth. To ensure uniform coverage of the bacterial lawn, bacterial cultures were spread evenly across the plate. The NGM plates +/- 3% L-arabinose were seeded with the bacteria and dried overnight. *E. coli*$^{Bt}$ and control strains, *E. coli*$^{WT}$ and *E. coli* OP50, were placed in airtight containers with anaerobic gas packs, and all plates were stored at 4˚C for one to two days prior to the experiment. Synchronized animals were plated and fed on the bacteria for 72 h (muscle polyQ and control) or 92 h (intestinal polyQ, neuronal polyQ and controls) prior to assessing aggregation or motility. Aggregates were quantified as described in "Aggregate Quantification". The same method was followed when using 3% glucose rather than 3% L-arabinose.

## Aggregate quantification

Worms were cultured and maintained as described in "*Caenorhabditis elegans* Maintenance and Strains". Following age synchronization, animals were plated on NGM plates seeded with specific bacteria. Unless otherwise indicated, after 92 h (intestinal polyQ44) or 72 h (muscle polyQ35) of feeding, animals were picked from the plates and placed in 96-well plates containing M9/levamisole, and frozen at -20˚C. Aggregates were quantified between 18–48 h post-freezing. PolyQ aggregates form rapidly, and their quantification is labor-intensive; therefore, the freezing step is essential to halt the formation of new aggregates during the quantification

process. Worms expressing intestinal polyQ (**Fig 1A and 1B**) were imaged using a Nikon Eclipse 80i epifluorescence microscope equipped with a 4X Plan Apochromat objective (0.2 NA), metal halide lamp, and Ex450-490nm/Em500-550nm excitation filter. For all other aggregate quantification assays, worms were viewed using a Zeiss Axiovert S100 equipped with an Achrostigmat 10X Ph1 phase-contrast infinity microscope objective (0.25 NA), Chroma EGFP/FITC long-pass filter set (19002), and a mercury lamp.

## Fecundity

Bacteria were grown as described in "Bacterial Culture Conditions". Synchronized L1s were plated on NGM containing bacteria with indicated concentrations of sodium butyrate and incubated for 48 h. After 48 h, worms were replated every 24 h until they were 96 h old. The number of adult worms per plate was accounted for after each transfer. To count the sum of the eggs and L1 larvae laid by gravid adults between 48–72 and 72–96 h of age, entire plates were washed with M9, pipetting up and down meticulously to ensure removal of eggs and larvae. Plates were verified void of eggs and larvae by confirming their removal under the Zeiss Stemi 305 stereo microscope. Eggs and larvae were washed twice with 15 mL M9 and were resuspended in 2 mL M9. Ten or 100 μL samples of M9 were taken from the falcon tube and eggs + L1-stage worms were counted in replicates of six per run. The total number of offspring was calculated by factoring in the dilution and was expressed as the total number of progenies per worm.

## RNAi knockdown

RNAi-mediated knockdown was carried out according to a previously published protocol [90]. Briefly, *E. coli* carrying DAF-16, SKN-1, and HSF-1 RNAi constructs and empty vector (L4440) were cultured overnight in a 37°C incubator shaking at 220 RPM in LB medium supplemented with 5 μg/mL tetracycline and 50 μg/mL ampicillin. Overnight cultures were induced for 4 h with 1 mM Isopropyl β-d-1-thiogalactopyranoside (IPTG) prior to seeding NGM plates supplemented with 12.5 μg/mL tetracycline, 100 μg/mL ampicillin, and 0.4 mM IPTG. Where specified, plates were also supplemented with 10, 25, 50, and 100 mM sodium butyrate. The seeded plates were allowed to dry overnight and were kept at 4°C for a maximum of two days. To bypass any effect on the development, L1 synchronized worms were placed on *E. coli* OP50 for 48 h at RT, washed three times with 15 mL M9, and plated on RNAi bacteria with or without butyrate. Aggregates were quantified after 48 h, as described under "Aggregate Quantification". Knockdown was confirmed using fluorescent reporters. The induction of *C12C8.1p*::GFP (AM446) was used to assess HSF-1 knockdown. The expression of *daf-16p*::DAF-16::GFP (TJ365) was used to assess DAF-16 knockdown. The induction of *gcs-1p*::GFP (LD1171) was used to assess SKN-1 knockdown. Each reporter strain was synchronized, plated on *E. coli* OP50 for 48 h, washed, and replated on corresponding RNAi plates. After 24 h on RNAi plates, worms expressing *C12C8.1p*::GFP were heat-shocked for 1 h at 33°C and allowed to recover on RNAi plates for additional 24 h before imaging. The strain expressing *daf-16p*::DAF-16::GFP was imaged after 48 h on RNAi. To induce the expression of *gcs-1p*::GFP, the RNAi plates seeded with bacteria expressing SKN-1 RNAi were supplemented with 5 mM acrylamide. Worms were imaged after 48 h on RNAi/acrylamide plates. For each strain, 10 worms were picked from control L4440 and test RNAi plate and mounted in 2 mM levamisole on 3% agarose pads covered with a coverslip. Images were taken using a Zeiss Axio Observer 7 microscope as described under "Live Imaging".

## Colonizing the *C. elegans* intestine with polyculture

All strains were grown as described in "Bacterial Culture Conditions", the cultures were normalized to $OD_{600}$ of 1.0, and were plated in equal parts on NGM plates +/- 3% L-arabinose. Worms were cultured as described in "*Caenorhabditis elegans* Maintenance and Strains", and aggregates were quantified as described in "Aggregate Quantification".

## Motility assays

All assays were performed at RT on age-synchronized animals. Unless otherwise indicated, worms were 72 h old at the time of body bend assays and 92 h old at the time of time-off-pick (TOP) assays. A body bend was defined as a change in the direction of the midline constituting approximately a 90˚ bend and a significant decrease in nose-to-tail distance. The rate was determined by placing animals in a drop of M9, allowing them to recover for 1 min, and counting the number of body bends in 30 s. TOP motility was defined by the time it took a worm to crawl completely off an eyebrow hair that was placed under its mid-section (**Fig 2A**). Worms were visualized using Zeiss Stemi 305 stereo microscopes.

## PolyQ aggregation in the progeny

Worms were cultured and maintained as described in "*Caenorhabditis elegans* Maintenance and Strains". Following synchronization, parental generation worms (P) were plated on NGM plates seeded with bacteria. After 72–78 h, animals (P) were bleached and the F1 progeny were age-synchronized, followed by plating on NGM plates containing control *E. coli* OP50 lawn. The F2 progeny were isolated by repeating the same method. F1 and F2 progeny were grown for 92 h at RT prior to aggregate quantification.

## Confirmation of bacterial killing after embryo isolation and age-synchronization of progeny

Worms were cultured and maintained as described in "PolyQ Aggregation in the Progeny". Following both the bleaching of the parental generations (P), and overnight synchronization of F1 L1s, 100 μL of M9 containing embryos and L1s, respectively, were spread across LB plates, put at 37˚C overnight, and examined the following day for the presence or absence of bacterial growth.

## Bacterial colonization assays

NGM plates were supplemented with indicated concentrations of sodium butyrate or 3% L-arabinose and seeded with bacteria. *E. coli* OP50 and *P. aeruginosa* PAO1 were grown under aerobic conditions and *E. coli*[Bt] was grown anaerobically. The plates were then dried. Worms expressing intestinal polyQ44 were age-synchronized and plated on NGM seeded with the above bacteria, followed by a 3- and 4-day incubation. To remove any extracellular bacteria, 12 or more worms from the plate of interest were transferred onto a plain NGM plate for approximately 30 min. Ten worms were then transferred from each plain NGM plate into 1 mL of M9 supplemented with 80 μM levamisole and the tube was agitated to move worms around in solution. Once worms settled to the bottom, 950 μL of the supernatant was gently removed by pipetting. To remove any remaining external bacteria, worms were washed twice, each wash lasting 10 min, with M9 containing gentamicin (100 μg/mL) and levamisole (80 μM). Worms were then washed three times with M9/levamisole and the last wash was saved. To account for any external bacteria, the last wash was plated onto LB plates. Two hundred microliters of 1% triton X-100 was added to the worms sitting in 50 μL M9/levamisole. A handheld homogenizer

(Bel-Art) was used to grind worms until no large remains were visible. The samples were then serially diluted with M9 and spotted onto LB plates. Plates were incubated at 37˚C overnight and colonies were quantified the next day.

## Butyrate dose response in axenic culture

Bacterial strains were cultured in LB. Strains were grown overnight in a 37˚C incubator shaking at 220 RPM. Two-fold serial dilutions of sodium butyrate were prepared in LB in a 96-well plate. The concentrations ranged from 0 to 200 mM. Three microliters of culture adjusted to $OD_{600}$ of 1.0 were added to 1.5 mL of LB media. An equal volume of this culture was added to the wells of the 96-well plate, which diluted the final concentrations of sodium butyrate by half (0 to 100 mM). The plate was incubated at 37˚C in a Tecan Infinite 200 Pro plate reader for 24 h. Every 30 min, the plate was shaken for 5 s and $OD_{600}$ readings were taken.

## Bacterial viability on NGM supplemented with butyrate

All bacteria were grown aerobically in LB overnight in a 37˚C incubator shaking at 220 RPM. The next day, cultures were adjusted to $OD_{600}$ of 0.5 and 10 μL of a $10^{-1}$ dilution were spotted onto NGM plates supplemented with 0, 10, 25, 50, or 100 mM sodium butyrate. The plates were then placed in a 37˚C incubator for 48 h. After incubation, the spots of bacteria were scraped with an inoculating loop, suspended in 1 mL LB, and 10 μL were spotted onto LB agar plates and placed in a 37˚C incubator overnight. Images were taken the next day and viability was assessed by the presence or absence of bacterial lawns or distinguishable colonies.

## Live imaging

Nematodes expressing intestinal polyQ33 or polyQ44 fed *E. coli* OP50 and *P. aeruginosa* PAO1 for four days were mounted in a drop of 1 mM levamisole on a 3% agarose pad and covered with a coverslip. GFP fluorescence and Nomarski images were taken using a Zeiss Axio Observer 7 microscope equipped with an Axiocam 503 mono camera, Solid-State Light Source Colibri 7, and a 5x Plan-Neofluar objective (0.16 NA). Pharyngeal mutant worms (DA597) were imaged using a 20x Plan-Neofluar objective (0.5 NA) and were processed using ZEN Tiles & Positions Module in Zeiss ZenPro software.

## Arabinose availability assay

Bacteria expressing the arabinose reporter (*E. coli* DC228) were inoculated with 2 μg/mL gentamicin and were cultured as described in "Bacterial Culture Conditions". Bacteria were seeded onto NGM plates +/- 3% L-arabinose and kept at RT for two days before worms were plated on them. Pharyngeal mutant animals (DA597) were age-synchronized and cultured for four days on *E. coli* bacteria (DC228) carrying an L-arabinose-inducible reporter (Ara-sfGFP) on NGM +/- 3% L-arabinose (positive and negative control, respectively). Animals used to test the availability of arabinose to the intestine were cultured for two days on *E. coli* bacteria carrying L-arabinose inducible reporter in the absence of L-arabinose followed by their transfer onto 3% L-arabinose plates containing *E. coli* OP50. Image acquisition is described in "Live Imaging".

## Western blotting and preparation

Worms were washed off NGM plates with M9, washed with 10 mL M9, and plated on unseeded NGM plates for ~30 min, or until M9 was dry to allow picking. To obtain soluble protein fractions, 20–50 (number was kept consistent between blots) worms were prepared

and samples were processed as previously described [91]. To obtain the insoluble fraction of polyQ44::YFP aggregates, 50 worms were picked in 10 μL M9 with 1 mM phenylmethylsulfonyl fluoride (PMSF) and were flash-frozen three times in -80 °C ethanol bath. One glass bead was placed in each tube and tubes were vortexed for 1 min and placed on ice for 1 min for a total of five cycles. Tubes were spun down 1500 RPM for 10 min. Nine microliters of samples were transferred into a new tube and combined with 9 μL 2x XT loading buffer and reducing agent. Samples were heated at 98 °C for 7 min. Twelve microliters of samples were loaded into wells. Proteins were separated on 4–12% gradient sodium dodecyl sulfate-polyacrylamide gel (SDS-PAGE) gels by electrophoresis and transferred onto polyvinylidene difluoride (PVDF) membrane using Trans-Blot Turbo Transfer machine in the presence of Trans-Blot Turbo Transfer Buffer. Membranes were blocked with 5% nonfat milk in 0.1% PBS-Tween-20 (PBST) for 30 min and probed with Living Colors JL-8 monoclonal primary antibody (1:2000, soluble fraction; 1:800, insoluble fraction) for at least 24 h at 4 °C followed by (1:40,000) goat-anti-mouse horse-radish peroxidase (HRP) conjugated secondary antibody. Clarity Western ECL substrate was used for chemiluminescence. Image J (v1.52) was used to quantify western blot bands.

## Quantification and statistical analysis

Data were considered statistically significant when p<0.05 was obtained by Student's t-test or one-way ANOVA followed by multiple comparison Dunnett's post-hoc test as indicated in figure legends. Asterisks denote corresponding statistical significance (*p<0.05, **p<0.01, ***p<0.001, ****p<0.0001). Where indicated, data are representative of the mean normalized to control strain *E. coli* OP50. Error bars represent standard error of the mean. Statistical analyses were performed using GraphPad Prism 8.4.3 software.

The numerical data used in all figures are included in S1 Data.

## Supporting information

**S1 Fig. Expression of polyQ44::YFP is not altered by bacteria or sodium butyrate in *C. elegans* intestinal polyQ44::YFP. A)** Western blot confirmation of antibody specificity. N2: no band; *gcs-1p*::GFP, polyQ33::YFP, polyQ44::YFP all show bands corresponding to their increasing molecular weight, respectively. **B-E)** Western blotting and image-J quantification of the soluble fraction of polyQ44::YFP in four-day-old *C. elegans* colonized with: **B)** *E. coli* OP50 and *P. aeruginosa* PAO1, **C)** *E. coli* OP50 with 0, 25, and 100 mM butyrate, **D)** *P. aeruginosa* PAO1 with 0, 25, 100 mM butyrate, **E)** F1 progeny from parental generations colonized with *E. coli* OP50 and *P. aeruginosa* PAO1. Band intensity is measured in arbitrary units (A.U). Data are representative of five independent experiments, **B**; four independent experiments, **C** and **D**; three independent experiments, **E**. Error bars represent SEM. Significance for **C** and **D** was calculated using one-way ANOVA followed by multiple comparison Dunnett's post-hoc test. Significance for **B** and **E** was calculated using Student's t-test (ns = not significant). (TIF)

**S2 Fig. Western blotting of polyQ44::YFP insoluble fractions confirms aggregation profiles in animals expressing intestinal polyQ44::YFP. A-C)** Insoluble fraction of polyQ44::YFP aggregates in four-day-old *C. elegans* expressing intestinal polyQ44::YFP colonized with: **A)** *E. coli* OP50 and *P. aeruginosa* PAO1, **B)** *E. coli* OP50 in the presence of 0, 25, 100 mM butyrate, and **C)** *P. aeruginosa* PAO1 in the presence of 0, 25, 100 mM butyrate. **D)** insoluble polyQ44 extracted from F1 progeny whose parents were colonized with *E. coli* OP50 or *P. aeruginosa*

PAO1.
(TIF)

**S3 Fig. Confirmation of the Time-Off-Pick (TOP) phenotype using established body bend readout. A)** Age-dependent decline in motility assessed by increased TOP (left) and decreased number of body bends per 30 seconds (right) in muscle-specific polyQ35 (AM140), muscle-specific polyQ40 (AM141), and control (N2) worms. The data are represented as the average TOP seconds or average number of body bends per worm normalized to day 3. Each data point represents the average of two independent experiments with a total of 30 worms. **B)** The effect of bacteria on the motility at day 3 assessed by TOP (left) and body bends (right) in muscle-specific polyQ35, muscle-specific polyQ40, and N2 control worms. Data are represented as the average TOP or average number of body bends per worm normalized to animals fed *E. coli* OP50. Each bar is an average of two independent experiments with a total of 30 worms. Error bars represent SEM. Significance was calculated using one-way ANOVA followed by multiple comparison Dunnett's post-hoc test ($^*$p<0.05, $^{****}$p<0.0001).
(TIF)

**S4 Fig. Colonization of worms expressing intestinal polyQ33 with *P. aeruginosa* does not induce aggregation.** Fluorescent images represent worms expressing either polyQ33 or polyQ44 that were fed control *E. coli* OP50 or test strain *P. aeruginosa* PAO1 for a period of four days. Scale bar = 200 μm.
(TIF)

**S5 Fig. F1 progeny/embryos are not exposed to bacteria during synchronization.** Images represent LB agar plates seeded with samples of M9 media from a 3$^{rd}$ wash of embryos post-bleaching and from overnight incubation of embryos that were allowed to hatch into L1 stage. Parental strains were colonized either with *E. coli* OP50 or *P. aeruginosa* PAO1. No colony-forming units were detected, indicating that all samples were void of bacteria.
(TIF)

**S6 Fig. The effect of butyrate supplementation on bacterial growth.** The growth of bacterial cultures was assessed in the presence of butyrate (0–100 mM) by measuring optical density at 600 nm ($OD_{600}$).
(TIF)

**S7 Fig. Bacterial viability on NGM supplemented with butyrate. A)** A cartoon depicting the procedure. **B)** Growth of bacteria collected from butyrate NGM plates and spotted on LB agar.
(TIF)

**S8 Fig. Comparison of the intestinal aggregation profiles of worms fed select bacteria on 100mM butyrate supplemented NGM vs. heat-killed (HK) bacteria.** Data are represented as the average number of aggregates per worm. Each bar is an average of three independent experiments with a total of 100 animals. Error bars represent SEM. Statistical significance was calculated using Student's t-test (ns: non-significant, $^{***}$p<0.0005, $^{****}$p<0.0001).
(TIF)

**S9 Fig. Enumeration of *E. coli* OP50 extracted from intestines of worms cultured on butyrate.** Bacterial load was enumerated by extracting intestinal *E. coli* OP50 on day four from animals expressing intestinal polyQ44 reporter (AM738). Data are represented as the average bacterial load per *C. elegans* intestine normalized to the control (0 mM butyrate). Each bar represents three independent experiments with a total of 30 animals. Error bars represent SEM. Significance was calculated using one-way ANOVA followed by multiple comparison

Dunnett's post-hoc test (ns: non-significant, ****p<0.0001).
(TIF)

**S10 Fig. The effect of butyrate on fecundity is dose- and polyQ-dependent.** The effect of 0, 25, 100 mM butyrate on fecundity, days three and four, in: **A)** N2 worms colonized with *E. coli* OP50, **B)** polyQ44 colonized with *E. coli* OP50, **C)** polyQ44 colonized with *P. aeruginosa* PAO1. Each bar represents the average of three (B) and two (A, C) independent experiments with a total of 93 and 62 worms, respectively. Error bars represent SEM. Significance was calculated using one-way ANOVA followed by multiple comparison Dunnett's post-hoc test (ns: non-significant, *p<0.05, ****p<0.0001).
(TIF)

**S11 Fig. Enumeration of bacteria in the *C. elegans* intestine.** Bacterial load was enumerated by extracting intestinal bacteria (*E. coli* OP50, *P. aeruginosa* PAO1, *E. coli*[Bt] +/- L-arabinose) on days three and four from animals expressing intestinal polyQ44 reporter (AM738). Each bar represents an average of three independent experiments with a total of 30 animals. Error bars represent SEM.
(TIF)

**S12 Fig. Glucose affects polyQ aggregation.** Data are represented as the average number of aggregates of intestine-specific polyQ44 (AM738) per worm normalized to the control (no glucose). Each bar is an average of two independent experiments with a total of 40 animals. Error bars represent SEM. Statistical significance between each pair was calculated using Student's t-test (**p<0.01, ***p<0.001, ****p<0.0001).
(TIF)

**S13 Fig. Availability of L-arabinose to bacteria harbored in the intestine assessed by the inducibility of L-arabinose fluorescent reporter. A)** An overlay of Nomarski and GFP images of *phm-2* worms cultured for four days on *E. coli* bacteria (DC228) carrying an L-arabinose-inducible reporter (Ara-sfGFP). **B)** *Phm-2* worms cultured on *E. coli* bacteria in the presence of 3% L-arabinose. **C)** *phm-2* worms cultured for two days on *E. coli* bacteria carrying L-arabinose inducible reporter in the absence of L-arabinose followed by a transfer onto 3% L-arabinose plates containing *E. coli* OP50. Scale bar = 200 µm.
(TIF)

**S1 Data. All numerical data.** Excel spreadsheet containing, in separate worksheets, the underlying numerical data for Figure panels: 1A, 1B, 1C, 1F, 1G, 2B, 2C, 3A, 3B, 3C, 3D, 3E, 3F, 3G, 3I, 4, 5, 6, 7, 8A, 8B, 9A, 9B, 9C, 9D, 9E, 10A, S1, S3A, S3B, S6, S8, S9, S10, S11, S12.
(XLSX)

## Acknowledgments

We thank Dr. Howard Shuman (University of Chicago), Dr. Keelnathan Shanmugam (University of Florida), Alice Prince (Columbia University), the University of Chicago Clinical Microbiology Laboratory, and the NorthShore Research Institute for sharing bacterial strains. Also, we want to thank Dr. Richard Morimoto and Sue Fox (Northwestern University) for generously providing *C. elegans* strains. The other *C. elegans* strains were provided by the Caenorhabditis Genetics Center, which is funded by NIH Office of Research Infrastructure Programs (P40 OD010440). We want to thank Dr. Brent Christner (University of Florida) and Dr. Mariola Edelmann (University of Florida) for allowing us to use their laboratory space and equipment and Dr. Keith Choe (University of Florida) for advice on experiments. Finally, we want

to thank Dr. Cindy Voisine, Mark Gorelik, and members of the Czyz Lab for reviewing our manuscript and providing insightful feedback. Cartoon figures were created using BioRender. com. The following reagents were obtained through BEI Resources, NIAID, NIH as part of the Human Microbiome Project: *Klebsiella oxytoca* Strain MIT 10–5243, HM-624; *Prevotella disiens* Strain DNF00882, HM-1171; *Prevotella corporis* Strain MJR7716 HM-1294.

## Author Contributions

**Conceptualization:** Alyssa C. Walker, Daniel M. Czyż.

**Formal analysis:** Alyssa C. Walker, Daniel M. Czyż.

**Funding acquisition:** Daniel M. Czyż.

**Investigation:** Alyssa C. Walker, Rohan Bhargava, Alfonso S. Vaziriyan-Sani, Christine Pourciau, Emily T. Donahue, Autumn S. Dove, Michael J. Gebhardt, Garrett L. Ellward, Daniel M. Czyż.

**Methodology:** Alyssa C. Walker, Tony Romeo, Daniel M. Czyż.

**Resources:** Tony Romeo.

**Supervision:** Daniel M. Czyż.

**Validation:** Alyssa C. Walker, Daniel M. Czyż.

**Visualization:** Alyssa C. Walker, Daniel M. Czyż.

**Writing – original draft:** Alyssa C. Walker, Daniel M. Czyż.

**Writing – review & editing:** Alyssa C. Walker, Rohan Bhargava, Alfonso S. Vaziriyan-Sani, Autumn S. Dove, Michael J. Gebhardt, Tony Romeo, Daniel M. Czyż.

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
