## [Decision Letter · Decision Letter 0]

12 Nov 2020

Dear Dr. Czyz,

Thank you very much for submitting your manuscript "Colonization of the Caenorhabditis elegans gut with human enteric bacterial pathogens leads to proteostasis disruption that is rescued by butyrate" for consideration at PLOS Pathogens. As with all papers reviewed by the journal, your manuscript was reviewed by members of the editorial board and by several independent reviewers. In light of the reviews (below this email), we would like to invite the resubmission of a significantly-revised version that takes into account the reviewers' comments.

We cannot make any decision about publication until we have seen the revised manuscript and your response to the reviewers' comments. Your revised manuscript is also likely to be sent to reviewers for further evaluation.

Sincerely,

Andreas J Baumler

Associate Editor

PLOS Pathogens

Brian Coombes

Section Editor

PLOS Pathogens

Kasturi Haldar

Editor-in-Chief

PLOS Pathogens

orcid.org/0000-0001-5065-158X

Michael Malim

Editor-in-Chief

PLOS Pathogens

orcid.org/0000-0002-7699-2064

Reviewer's Responses to Questions

**Part I - Summary**

Reviewer #1: The human microbiome is suspected of influencing morbidity and mortality due to protein conformational diseases such as Alzheimer’s, Parkinson’s and Lou Gehrig’s diseases. However, the complexity of the human gut and inability to precisely control experimental conditions therein have hindered progress understanding the mechanisms underpinning these phenomena. Modeling microbiome connections to organismal protein homeostasis (proteostasis) has been successful in the nematode C. elegans, where ingestion of human pathogenic bacteria has been demonstrated to cause proteostatic collapse. In this report, the authors utilize the worm intestinal colonization approach paired with sensitive poly-glutamine-GFP proteostasis reporters to demonstrate that several important human pathogens disrupt proteostasis both immediately in the gut as well as in distal tissues including muscle and neuron. The short chain fatty acid butyrate has been shown to partially ameliorate such phenotypes in other studies, and indeed the same occurs here – exogenous or bacterially produced butyrate suppressed both polyQ aggregation and resultant phenotypes.

The work presented in this report is cleanly executed and the results largely clear and compelling. It teeters on the edge of incrementalism given the wealth of reports preceeding it on the topic, but breaks new ground with the butyrate experiments that clearly tie pathogen-induced proteostasis defects with previous findings. Although not fully explored, the observation that pathogen-induced proteostasis collapse extends into the F1 generation is striking. Additionally, the use of several pathogenic bacteria strongly reinforce the observations and lend credence to the model that pathogens disrupt protein quality control not just in local but also distal tissues, with profound ramifications for human biology. The only thing lacking is mechanistic insight into how butyrate is performing the observed rescues – more on this below. Overall this should be an important addition to the proteostasis, worm and neurodegeneration communities.

Reviewer #2: In this paper “Colonization of the Caenorhabditis elegans gut with human enteric bacterial pathogens leads to proteostasis disruption that is rescued by butyrate”, Czyz and colleagues evaluate the effects of exposing a panel of C. elegans strains expressing polyQ expansion proteins in the gut, muscle and neurons, to a panel of Gram-negative enteric bacterial pathogens. The goal of their studies is to understand the effects of the gut microbiome on protein aggregation and toxicity. The authors show that a diet of pathogenic bacterial strains predominantly exacerbates polyQ aggregation and impairs mobility of the animal, while exposing animals to bacteria that conditionally synthesize butyrate, suppresses this detrimental effect. The data are intriguing and suggestive that gut bacteria can influence protein aggregation not only in the gut, but also indirectly, the aggregation of protein in other tissues. However, the conclusions of the manuscript are still rather premature and descriptive, the non-autonomous effects described are not substantiated and could be due to less interesting reasons such as the health of the animals, and there are technical and conceptual issues which this reviewer thinks the authors should address before the paper is acceptable for publication.

Reviewer #3: Manuscript by Walker et al investigates how the intestinal colonization of C. elegans by enteric pathogens affect protein folding in the intestine and other systemic sites. Authors show that the colonization of the C. elegans intestine with enteric bacterial pathogens disrupts organismal proteostasis and enhances aggregation of polyQ and decrease motility. They use animals that constitutively express intestine-specific polyQs fused to yellow fluorescent protein (polyQ44::YFP) to quantify aggregation. They also use similar models to assess protein aggregation in muscles, neurons and gonads. Authors also show that butyrate can reverse polyQ aggregation and proteotoxicity and improve host proteostasis. Overall, this is a very interesting study and important for the understanding of how microbiota or pathogens can influence protein aggregation in the gut. There are some controls missing in the figures. I would highly recommend identifying whether the effect of enteric bacteria are inflammation driven or not. This could be achieved by comparing the commensal strains to pathogenic strains of enteric bacteria, or using sterile inflammation or avirulent mutants. I would also highly recommend testing the effect of butyrate on enteric bacteria as recommended below. Finally, recent studies suggested that curli expressing enteric bacteria can affect alpha synuclein aggregation. It would improve the impact of the manuscript if authors can address whether enteric curli mediates the polyQ aggregation in the gut.

**Part II – Major Issues: Key Experiments Required for Acceptance**

Reviewer #1: 1. Previous work in worms and elsewhere has tied butyrate to inhibition of histone deacetylases and therefore transcriptional stress responses that may counteract the observed increase in protein aggregates. Given the tractability of the worm model and the ease of doing the experiments, it seems a missed opportunity for the authors to not try and connect the dots. I strongly suggest using DAF-16, SKN-1 and HSF-1 mutant worm lines to repeat the butyrate rescue experiments to determine whether one or more of these well-established proteostasis-supporting pathways are involved.

2. Fig. 8 is a little perplexing to this reviewer. The butyrate-producing E. coli appear to reduce aggregates in the control strain as much or more than they reduce it in the presence of P. aeruginosa. One interpretation is that the butyrate effect has nothing to do with the pathogen presence per se and is simply a physiological modulator of proteostasis in general. I suspect if these two experiments weren’t normalized, the aeruginosa effect would look much more dramatic, since the control strain should have few aggregates to start with.

Reviewer #2: 1) The mechanisms by which pathogenic bacteria cause an increase in protein aggregation are not adequately addressed. Are the animals dying earlier, sick or nutrient deprived, or is the innate immune response activated and thus causing an increase in ROS-induced damage, etc.? These questions need to, at the very least, be addressed in order to evaluate the significance or novelty of the tissue-nonautonomous influence (ln 137) that the authors highlight. If it is simply that intestinal cells are dysfunction, and therefore muscle cells are also nutrient deprived, it would be suggestive of one, perhaps less exciting mechanism. On the other hand, if the effects are due to hormonal signaling between the intestine and muscle, it would be suggestive of another, perhaps more intriguing mechanism.

2) The expression levels of polyQ under the different treatments should be measured to assess whether changes in protein expression explain the observed phenotypes.

3) The methodology associated with determining mobility is worrisome. The authors have devised a creative, and perhaps useful new method to assess mobility in animals harboring a roller phenotype (the TOP assay). However, no controls are included to address what is being measured using the TOP assay. Is the ability of animals to slide off an eyebrow hair a reflection of muscle contractility or some function of cuticle texture, hydrostatic pressure, eggs in the animal’s uterus, etc.? Controls using mutants known to be defective in some of these physiological features could provide more credence to the assay.

4) The observation that the F1 progeny from infected mothers have higher aggregate loads is intriguing. Is protein synthesis altered in these animals? Do these animals express more polyQ proteins? Does bleach definitively remove all bacteria that the parents were in contact with?

5) It is known that decreasing fecundity has profound effects on polyQ aggregation and toxicity. Does butyrate affect fecundity? If so, depending on the mechanism, this may diminish the novelty of the observation.

6) In Figure 7, the authors should provide some way of rationalizing why polyQ animals fed E. coli expressing butyrate (E. coli Bt + arabinose) move more than wild-type animals exposed to the same conditions.

7) Are developmental rates the same under all conditions?

8) This is applicable to multiple experiments: given the number of treatments, statistical analysis should be corrected for multiple comparisons (t-tests between two treatments alone can be misleading).

Reviewer #3: 1-Can the commensal enteric bacteria influencing polyQ aggregation similar to pathogenic enteric strains? The authors can test this by comparing pathogenic versus commensal strains of couple of the bacteria tested here. It would tell us whether bacterial virulence mechanisms contribute to the polyQ aggregation. Alternatively, non-pathogenic or avirulent mutants of the bacterial strains can be used to as the same question.

2-Can the authors elaborate more on the effect of butyrate? It would be a good addition to the manuscript at least to show if the same effect observed by bacteria can be recapitulated by a sterile inflammation by using low dose DSS? Earlier studies have shown that the supplementation of butyrate suppresses enteric bacterial expansion in the gut. This could be a similar mechanism here that the authors can show that the reversal of the inflammation may be leading to the reduction of the aggregation. If the authors can not use a chemical agent similar to DSS maybe they can plate the bacteria in Fig 8 and show that the numbers of enterics go down in the presence of butyrate.

3-Although the effect of microbiota on host proteostasis could be attributed to systemic inflammation, recent studies have shown that the curli amyloid secreted by E. coli and other enteric bacteria can directly seed alpha synuclein deposition in mouse model of PD (Sampson et al 2020 eLife). Can authors comment on whether curli expressed in the C. elegans gut can effect the proteostasis in their model system? I would highly recommend adding an E. coli curli + and – or a S. Typhimurium curli+ and – strains to their assays to improve this knowledge gap and increase the impact of the paper.

**Part III – Minor Issues: Editorial and Data Presentation Modifications**

Reviewer #1: Minor issues:

1. Suggest that the titles of figure panels be removed and only stated in the legends. The figure layout resembles a slide presentation more than a figure set at this point.

2. The polyQ reporters differ depending on the tissue being interrogated. While clear to aficionados, the reasoning behind these choices is not presented in the text. For example, Q33 is a control in Fig. 1, and Q35 an experiment in Fig. 2. A little more background would help.

Reviewer #2: 9) To show that arabinose causes the expression of butyrate the authors have used a phm-2 mutant strain. It is my understanding that the remaining experiments were not conducted in this background. The authors should explain the rationale for this difference, and also whether this difference impacts their remains experiments.

Reviewer #3: Fig 1A. No bacteria control is missing. It would be informative to see the kinetics of aggregation without E. coli. It is hard to say if there is any accelerated aggregation.

Fig1B. Can the authors comment on the species that didn’t show any aggregation? This information is not included in the results section. It would be informative to know if this strain is a commensal strain where as the others are pathogenic.

To draw the conclusion that the polyQ aggregation is necessary for the pathogen-induced effect, the authors should repeat the aggregation experiment both in PolyQ33 and polyQ44 background. This wwould support the results obtained in Fig 1E .

Fig 3. Have the authors considered plating the bacteria straight from the nematodes to see if butyrate effects the in vivo numbers? It is possible that the effect of the butyrate maybe stronger in vivo due to additional factors compared to in vitro treatment.

Fig 3. Does butyrate treatment effect aggregation without the bacteria? No bacteria controls are missing.

Fig 4. Does butyrate treatment effect motility without the bacteria? No bacteria controls are missing.

PLOS authors have the option to publish the peer review history of their article (what does this mean?). If published, this will include your full peer review and any attached files.

Reviewer #1: **Yes: **Kevin Morano

Reviewer #2: No

Reviewer #3: No
---

## [Editor Report · Decision Letter 1]

29 Mar 2021

Dear Dr. Czyz,

We are pleased to inform you that your manuscript 'Colonization of the Caenorhabditis elegans gut with human enteric bacterial pathogens leads to proteostasis disruption that is rescued by butyrate' has been provisionally accepted for publication in PLOS Pathogens.

Best regards,

Andreas J Baumler

Associate Editor

PLOS Pathogens

Brian Coombes

Section Editor

PLOS Pathogens

Kasturi Haldar

Editor-in-Chief

PLOS Pathogens

orcid.org/0000-0001-5065-158X

Michael Malim

Editor-in-Chief

PLOS Pathogens

orcid.org/0000-0002-7699-2064
---

## [Editor Report · Acceptance letter]

14 Apr 2021

Dear Dr. Czyz,

We are delighted to inform you that your manuscript, "Colonization of the Caenorhabditis elegans gut with human enteric bacterial pathogens leads to proteostasis disruption that is rescued by butyrate," has been formally accepted for publication in PLOS Pathogens.

Best regards,

Kasturi Haldar

Editor-in-Chief

PLOS Pathogens

orcid.org/0000-0001-5065-158X

Michael Malim

Editor-in-Chief

PLOS Pathogens

orcid.org/0000-0002-7699-2064